# On the Convergence of Hierarchical Federated Learning with Partial Worker Participation

**Xiaohan Jiang**[1,2]          **Hongbin Zhu**[2]

[1]School of Computer Science, Fudan University, Shanghai, China
[2] Institute of FinTech, Fudan University, Shanghai, China

## Abstract

Hierarchical federated learning (HFL) has emerged as the architecture of choice for multi-level communication networks, mainly because of its data privacy protection and low communication cost. However, existing studies on the convergence analysis for HFL are limited to the assumptions of full worker participation and/or i.i.d. datasets across workers, both of which rarely hold in practice. Motivated by this, we in this work propose a unified convergence analysis framework for HFL covering both full and partial worker participation with non-i.i.d. data, non-convex objective function and stochastic gradient. We correspondingly develop a three-sided learning rates algorithm to mitigate data divergences issue, thereby realizing better convergence performance. Our theoretical results provide key insights of why partial participation of HFL is beneficial in significantly reducing the data divergences compared to standard FL. Besides, the convergence analysis allows certain individualization for each cluster in HFL indicating that adjusting the worker sampling ratio and round period can improve the convergence behavior.

## 1 INTRODUCTION

Federated Learning (FL) [McMahan et al., 2017, Yang et al., 2019] is a privacy-preserving machine learning paradigm for substantial decentralized data. FL allows a large number of workers to collaboratively learn a model with their local data under the coordination of a centralized server. Formally, the goal of FL is to solve an optimization problem, which is

$$\min_{\mathbf{x} \in \mathbb{R}^d} f(\mathbf{x}) := \frac{1}{m} \sum_{i=1}^{m} F_i(\mathbf{x}), \qquad (1)$$

where $F_i(\mathbf{x})$ is the local (non-convex) loss function parameterized by $\mathbf{x}$ and $m$ is the number of workers in total. We can tackle the above problem iteratively in a distributed way, where most representative algorithm is FedAvg [McMahan et al., 2017]. Specifically, the workers each train on their local data, take several gradient steps, and then forward their locally updated model to the server to be averaged. This eliminates the need for explicitly sharing sensitive data with others, thereby providing privacy guarantee. A large body of work demonstrates the benefits of FedAvg via both theoretical convergence analysis and empirical experiments [Li et al., 2019, Wang and Joshi, 2021].

On the other hand, standard FL setting with single cloud is inapplicable to substantial real world scenarios with low latency requirement. Multi-level network architecture enables great potential in low latency FL, such as edge computing. Specifically, edge computing allows worker to transmit the updated model to local server in the vicinity, thereby significantly reducing the transmission latency [Zhang et al., 2022]. Local severs collect the updated models and send them to the cloud sever. Besides, multi-level network boosts the number of connected end devices by providing more access points in the edge.

To accommodate the multi-level network architecture, a few works proposed hierarchical FL (HFL). In HFL, workers are partitioned into multiple groups, with each group governed by a cluster (local server), and all clusters coordinated through a master (global server). Specifically, after several local iterations, workers first send their updated models to cluster for local aggregation within the belonging cluster. After several local aggregations, all clusters communicate with the master for global aggregation among groups. This two-level aggregation in HFL strikes a subtle balance between the communication overhead and learning performance. Parallel local aggregations ensure ultra-low latency, while the time-consuming global aggregation embraces extensive model knowledge.

Very recent, there have been a few works analyzing the

Table 1: A summary of convergence rates of optimization methods for HFL.

| Algorithm | SGD | Non-i.i.d. | Partial Worker | Convergence Rate[1] |
|---|---|---|---|---|
| Liu et al. [2020][2] | × | ✓ | × | $\mathcal{O}(\frac{B^G}{\sqrt{mT}})$ |
| Castiglia et al. [2020][2] | ✓ | × | × | $\mathcal{O}(\frac{1}{\sqrt{mT}} + \frac{mG^2}{IT})$ |
| Wang et al. [2022][2] | ✓ | ✓ | × | $\mathcal{O}(\frac{1}{\sqrt{mT}} + \frac{(M-1)G^2+(m-M)I^2}{T})$ |
| Liu et al. [2022a][3] | ✓ | × | × | $\mathcal{O}(\frac{1}{\sqrt{GT}} + \frac{M}{mT})$ |
| Ours[3] | ✓ | ✓ | ✓ | $\mathcal{O}(\frac{1}{\sqrt{mGT}} + \frac{1}{T})$ |

[1] $G$: global aggregation period (master period); $I$: local aggregation period (cluster period); $m$: number of workers in HFL system; $M$: number of groups; $B$ is a constant and $B > 2$.

[2] $T$ in these works refer to total number of local iterations.

[3] $T$ in these works refer to total number of master rounds.

convergence behavior under HFL scenario. In particular, Castiglia et al. [2020] considers a two-level FL where the clusters are organized as a peer-to-peer network. However, their theoretical results only cover the case of identically and independently distributed (i.i.d.) data. Liu et al. [2020] considers non-i.i.d. data, but they assume full (non-stochastic) gradients and the convergence bound is in an exponential function form. Wang et al. [2022] provides a more systematic analysis for HFL, which considers non-convex objective function, non-i.i.d. data, and stochastic gradient descent (SGD). They split the overall data heterogeneity (global divergence) into two components, worker-cluster divergence (i.e., within a group) and cluster-master divergence (i.e., among groups). Compared to standard FL, their results show that local aggregation can help to overcome worker-cluster divergence.

However, all the aforementioned works only consider full worker participation (FWP), which is often not possible in practice. Workers may randomly join or leave the FL system, making the active worker set stochastic and time-varying across communication rounds [Yang et al., 2021]. Waiting for all workers' responses can significantly slow down the training performance, especially when there are inactive workers or stragglers. This necessitates us to consider partial worker participation (PWP), where only a subset of the workers are chosen in each communication round. This is especially critical in HFL scenarios, such as edge computing, where workers may experience availability issues due to battery level, network status, incoming calls, etc. Therefore, there is a need for a comprehensive analysis and understanding of HFL with PWP.

In this paper, we present a novel theoretical analysis for HFL with both FWP and PWP. We follow the setting of Wang et al. [2022] for HFL, and newly derive the convergence bound for HFL w.r.t. communication rounds. This bound is more explicit than their original one [Wang et al., 2022] which was w.r.t. local iterations. Besides, we develop a generalized FedAvg with three-sided learning rates for HFL correspondingly, showing how local aggregation can help to overcome worker-cluster divergence and how to realize

linear speedup.

Our theoretical results reveal that PWP can facilitate HFL more effectively than standard FL. Specifically, HFL alleviates the additional uncertainty caused by PWP in terms of both worker-cluster and cluster-master divergences. This weakening effect yet is only observed on worker-cluster divergences in HFL with FWP. Therefore, enabling PWP is especially beneficial when certain data heterogeneity exists among groups, which is the most general case of HFL in practice. This suggests a mutual beneficial relationship between HFL and PWP. Many real world scenarios of HFL employ PWP to deal with the instability and unavailability of workers, such as edge computing. While exactly, the hierarchical architecture just provides a suitable showcase for PWP. Besides, our theoretical results show that HFL can be customized to a certain degree. Specifically, each cluster can adjust its worker sampling ratio and round period accordingly to deal with its inner data heterogeneity. A summary of our result with existing results is shown in Table 1.

We emphasize that our main contribution is to provide a new unified convergence bound for HFL settings. Our theoretical results recover the previous results of FL from Yang et al. [2021] by setting $M = 1, I = G$, and $\epsilon = 0$, and generalize existing HFL results to PWP and heterogeneous clusters $I_i$ and $n_i$. Compared to Wang et al. [2022], our work adopts the same typical HFL settings, but the convergence analysis framework is completely different. Specifically, we rederive a round-level convergence bound for both FWP and PWP, which is potentially tighter (in the sense of second term, our $\mathcal{O}(\frac{1}{T})$ versus their $\mathcal{O}(\frac{G}{T})$) and more general (aware of some server-sided optimization) than their iteration-level results. Compared to Yang et al. [2021], our work shares a similar algorithmic approach but differs in the analysis method. We do face several theoretical challenges specialized in HFL scenario, which has not been considered in existing FL works. We defer some details on this to Section 4.

We highlight our contributions as follows.

1. We derive a general convergence bound of HFL with both FWP and PWP, non-i.i.d. data, non-convexity,

and SGD. We correspondingly develop a three-sided learning rates algorithm.

2. Compared to standard FL, we reveal that PWP can significantly reduce both worker-cluster divergences and cluster-master divergence in HFL.

3. We provide certain individualization for HFL by suggesting each cluster flexibly set its worker sampling ratio and round period to match its inner divergence for potentially better convergence behavior.

4. We introduce comprehensive and reproducible empirical baselines for comparison. We conduct extensive numerical experiments on multiple datasets to verify our theoretical results.

## 2 RELATED WORKS

A large body work studied the convergence behavior of the FedAvg algorithm for standard FL. Some works focus on convex objective functions [Stich, 2019, Wang et al., 2019, Li et al., 2019], while others consider non-convex objective functions [Haddadpour et al., 2019, Yu et al., 2019b, Wang and Joshi, 2021]. There are also works that extend the theoretical results to non-i.i.d. data case [Li et al., 2019, Khaled et al., 2019]. Besides, several variants of FedAvg are proposed and analyzed, such as those that address system heterogeneity [Li et al., 2020], combine with momentum [Yu et al., 2019a], or use adaptive optimizers [Reddi et al., 2020].

For HFL, there have been a few works providing convergence analysis. These include Castiglia et al. [2020], Zhou and Cong [2019] for i.i.d. data, and Liu et al. [2020] for non-i.i.d. data with full gradient descent. Wang et al. [2022] provides a comprehensive analysis framework for HFL, covering non-convex objective function, non-i.i.d. data, and SGD. Several variants are also proposed and theoretically studied, with quantization technique [Liu et al., 2022a], momentum acceleration [Yang et al., 2023], over-the-air setup [Aygün et al., 2022], wireless resource allocation [Liu et al., 2022b], user mobility [Feng et al., 2022], data offloading [Ganguly et al., 2023], and submodel partitioning [Fang et al., 2023]. However, all these works assume FWP, and some even require stronger assumptions such as bounded gradient. Besides, there are also works on system design for HFL without convergence guarantees [Luo et al., 2020, Abad et al., 2020, Briggs et al., 2020].

For PWP, a popular branch of works focuses on the uniform worker sampling pattern with or without replacement. Some typical representatives include, for strongly convex objective function [Li et al., 2019], with proximal term to handle heterogeneity [Li et al., 2020], with extra communications to reduce variance introduced by PWP [Karimireddy et al., 2020]. Then Yang et al. [2021] improves the theoretical results and achieves a linear speedup with two-sided learning

rates. There are further some works addressing variance-reducing in PWP via memorized gradients [Jhunjhunwala et al., 2022] and momentum-based update [Das et al., 2022]. The work by Qu et al. [2022] considers a multi-server FL with overlapping area setting, also taking into account the uniform PWP. However, their multi-server scenario differs from HFL in that global aggregation never takes place in the training process. Some other works allow for arbitrary worker sampling probabilities and provide convergence analysis [Gu et al., 2021, Perazzone et al., 2022, Fraboni et al., 2023]. However, these works require much stronger assumptions, such as Lipschitz Hessian or/and bounded gradient assumption. Besides, another branch of works considers arbitrarily asynchronous participation patterns Avdiukhin and Kasiviswanathan [2021], Yang et al. [2022], Nguyen et al. [2022], Wang and Ji [2022]. In this paper, we base our work on uniform sampling with standard assumptions, as we will elaborate in Section 3 and Section 4.

## 3 HFL SETUP

Suppose there are $m$ in total workers making up a set $\mathcal{V}$. In two-level HFL setting, all workers are grouped into $M$ clusters $\mathcal{V}_1, \mathcal{V}_2, \ldots, \mathcal{V}_M$. Let $m_i := |\mathcal{V}_i|$ $(i = 1, 2, \ldots, M)$ denote the number of workers in each group cluster. Therefore, we have $m = \sum_{i=1}^{M} m_i$. With the cluster policy, the objective function of Eq. 1 is equivalent to

$$\min_{\mathbf{x} \in \mathbb{R}^d} f(\mathbf{x}) := \sum_{i=1}^{M} \frac{m_i}{m} f_i(\mathbf{x}),$$

where $f_i(\cdot)$ is the averaged loss function of workers in cluster $i$ which is

$$f_i(\mathbf{x}) := \frac{1}{m_i} \sum_{j \in \mathcal{V}_i} F_j(\mathbf{x}).$$

In HFL, workers conduct multiple SGD iterations to minimize the local objective function $F_j(\mathbf{x}) \triangleq \mathbb{E}_{\xi_j \sim D_j}[F_j(\mathbf{x}, \xi_j)]$ w.r.t. model parameters $\mathbf{x}$ on their own dataset $\mathcal{D}_j$. Each cluster $i$ $(i = 1, 2, \ldots, M)$ first averages the updated parameters from its workers $j \in \mathcal{V}_i$ every $I_i$ local iterations (refer to as a cluster round with period $I_i$). After several rounds of intra-cluster aggregations, a master globally averages the models from all $M$ clusters. This global aggregation takes place for every $G$ local iterations (refer to as a master round with period $G$). Note that $G$ is a common multiple of $\{I_1, I_2, \ldots, I_M\}$. Different $I_i$ values account for potential system heterogeneity (computation and communication capacity) of devices in different cluster.

We investigate a three-sided learning rates hierarchical FedAvg, which is essentially a generalization of previous works [Karimireddy et al., 2020, Reddi et al., 2020, Yang et al., 2021]. The algorithm is shown in Algorithm 1. For a natural number $m$, we use $[m]$ to represent the set

**Algorithm 1** HFL with Three-sided Learning Rates

---

1: **Input:** $\eta, \eta_c, \eta_g, \mathbf{x}^0, \{\mathcal{V}_i : i \in [M]\}, G, \{I_i : i \in [M]\}, \{\omega_i : i \in [M]\}, \{n_i : i \in [M]\}$.
2: **Output:** Global aggregated model $\bar{\mathbf{x}}^T$.
3: **for** $t = 0$ *to* $T - 1$ **do**
4:   **for** *each cluster* $i \in [M]$ *in parallel* **do**
5:     **for** $\tau = 0$ *to* $\omega_i - 1$ **do**
6:       Cluster $i$ samples a subset $\mathcal{S}_i^{t,\tau}$ of workers with $|\mathcal{S}_i^{t,\tau}| = n_i$.
7:       **for** *each worker* $j \in \mathcal{S}_i^{t,\tau}$ *in parallel* **do**
8:         **for** $h = 0$ *to* $I_i - 1$ **do**
9:           Compute a gradient estimate $\mathbf{g}_j^{t,\tau,h}$.
10:           Worker update: $\mathbf{x}_j^{t,\tau,h+1} = \mathbf{x}_j^{t,\tau,h} - \eta \mathbf{g}_j^{t,\tau,h}$.
11:         **end for**
12:         Let $\tilde{\Delta}_j^{t,\tau} = \mathbf{x}_j^{t,\tau,0} - \mathbf{x}_j^{t,\tau,I_i} = \eta \sum_{h=0}^{I_i-1} \mathbf{g}_j^{t,\tau,h}$
13:         Send $\tilde{\Delta}_j^{t,\tau}$ to cluster $i$.
14:       **end for**
15:       Cluster $i$ receives $\tilde{\Delta}_j^{t,\tau}, j \in \mathcal{S}_i^{t,\tau}$.
16:       Let $\Delta_i^{t,\tau} = \frac{1}{n_i} \sum_{j \in \mathcal{S}_i^{t,\tau}} \tilde{\Delta}_j^{t,\tau}$.
17:       Cluster update: $\bar{\mathbf{x}}_i^{t,\tau+1} = \bar{\mathbf{x}}_i^{t,\tau} - \eta_c \Delta_i^{t,\tau}$.
18:       Broadcast $\bar{\mathbf{x}}_i^{t,\tau+1}$ to workers in cluster $i$.
19:     **end for**
20:     Let $\Delta_i^t = \bar{\mathbf{x}}_i^{t,0} - \bar{\mathbf{x}}_i^{t,\omega_i} = \eta_c \sum_{\tau=0}^{\omega_i-1} \Delta_i^{t,\tau}$.
21:     Send $\Delta_i^t$ to master.
22:   **end for**
23:   Master receives $\Delta_i^t, i \in [M]$.
24:   Let $\Delta^t = \sum_i^M \frac{m_i}{m} \Delta_i^t$.
25:   Master update: $\bar{\mathbf{x}}^{t+1} = \bar{\mathbf{x}}^t - \eta_g \Delta^t$.
26:   Broadcast $\bar{\mathbf{x}}^{t+1}$ to all workers.
27: **end for**

---

$\{1, 2, \ldots, m\}$. The Learning rates for worker, cluster, and master are $\eta$, $\eta_c$, and $\eta_g$, respectively. Note that $h$, $\tau$, and $t$ always count for local iteration, cluster round, and master round, respectively. We let $\omega_i = \frac{G}{I_i}$ denote the number of cluster rounds for cluster $i, \forall i \in [M]$ in a master round. We denote the stochastic gradient estimator as $\mathbf{g}_j^{t,\tau,h} = \nabla F_j(\mathbf{x}_j^{t,\tau,h}, \xi_j^{t,\tau,h})$, where $\xi_j^{t,\tau,h}$ is the random data samples from the local dataset $\mathcal{D}_j$ at worker $j$ for iteration $h$ (in cluster round $\tau$ and master round $t$). For a cluster round $\tau$, $\tilde{\Delta}_j^{t,\tau}$ is the accumulated gradients of worker $j$, while $\Delta_i^{t,\tau}$ is the averaged gradients of all participated workers in cluster $i$.

In HFL with PWP, each cluster round only includes a certain subset of workers. We denote $\mathcal{S}_i^{t,\tau}$ as participating worker index set, which is determined once a new cluster round $\tau$ starts. We have $|\mathcal{S}_i^{t,\tau}| = n_i$, for some $n_i \in (0, m_i]$. For the sampling strategy of participating set, we employ two strategies proposed by Li et al. [2020] and Li et al. [2019], respectively. Specifically, we select $\mathcal{S}_i^{t,\tau}$ randomly and independently, either with replacement (Strategy 1) or without replacement (Strategy 2). For each member in $\mathcal{S}_i^{t,\tau}$, we

pick a worker from $\mathcal{V}_i$ uniformly at random with probability $p_j = \frac{1}{m_i}, \forall j \in \mathcal{V}_i$. Ultimately, the participation likelihood for any worker $j \in \mathcal{S}_i^{t,\tau}$ equals to $\frac{n_i}{m_i}$. We denote the total number of worker sampling size in HFL system as $n = \sum_{i=1}^M n_i$.

# 4 CONVERGENCE ANALYSIS

To establish the convergence theorem, we preset the following assumptions.

**Assumption 1** (L-Lipschitz Continuous Gradient). There exists a constant $L > 0$, such that $||\nabla F_i(\mathbf{x}) - \nabla F_i(\mathbf{y})|| \le L||\mathbf{x} - \mathbf{y}||, \forall i, \mathbf{x}, \mathbf{y}$.

Note that Lipschitz continuous gradient assumption internally applies to the cluster objective $f_i(\mathbf{x})$ and global objective $f(\mathbf{x})$. For instance, $||\nabla f_i(\mathbf{x}) - \nabla f_i(\mathbf{y})|| = ||\frac{1}{m_i} \sum_{j \in \mathcal{V}_i} \nabla F_j(\mathbf{x}) - \frac{1}{m_i} \sum_{j \in \mathcal{V}_i} \nabla F_j(\mathbf{y})|| \le \frac{1}{m_i} \sum_{j \in \mathcal{V}_i} ||\nabla F_j(\mathbf{x}) - \nabla F_j(\mathbf{y})|| \le L||\mathbf{x} - \mathbf{y}||$.

**Assumption 2** (Unbiased Local Gradient Estimator). Let $\xi_i^h$ be a random local data sample in the $h$-th step at the $i$-th worker. The local gradient estimator is unbiased, i.e., $\mathbb{E}[\nabla F_i(\mathbf{x}, \xi_i^h)] = \nabla F_i(\mathbf{x}), \forall i, \mathbf{x}$, where the expectation is over all local datasets samples.

**Assumption 3** (Bounded Variance). There exists a constant $\sigma > 0$, such that the variance of each local gradient estimator is bounded by $\mathbb{E}[||\nabla F_i(\mathbf{x}, \xi_i^h) - \nabla F_i(\mathbf{x})||^2] \le \sigma^2, \forall i, \mathbf{x}$.

**Assumption 4** (Bounded Cluster-Master and Worker-Cluster Divergence). The bounded cluster-master divergence is expressed as $||\nabla f_i(\mathbf{x}) - \nabla f(\mathbf{x})||^2 \le \epsilon^2, \forall i \in [M], \mathbf{x}$, while the bounded worker-cluster divergence is $||\nabla f_j(\mathbf{x}) - \nabla F_i(\mathbf{x})||^2 \le \epsilon_i^2, \forall i \in [M], j \in \mathcal{V}_i, \mathbf{x}$.

The first three assumptions are standard in non-convex optimization [Ghadimi and Lan, 2013, Bottou et al., 2018]. Assumption 4 quantifies the heterogeneity of the non-i.i.d. datasets among different workers and groups. It was first introduced by Wang et al. [2022] for HFL. The worker-cluster part measures the data heterogeneity among workers inside a group, while the cluster-master part measures the data heterogeneity among groups. In particular, $\epsilon^2 = 0$ stands for inter-group i.i.d., and $\epsilon_i^2 = 0$ for intra-group-$i$ i.i.d., respectively. In standard FL case, Assumption 4 has a simpler form as $||\nabla F_j(\mathbf{x}) - \nabla f(\mathbf{x})||^2 \le \tilde{\epsilon}^2, \forall j \in [m], \mathbf{x}$, which is also referred to as bounded global divergence [Yang et al., 2021, Reddi et al., 2020, Wang et al., 2019, Yu et al., 2019b]. Note that the worker-cluster and cluster-master divergences are essentially two components of the global divergence, since the total variance is $\frac{1}{m} \sum_{j=1}^m ||\nabla F_j(\mathbf{x}) - \nabla f(\mathbf{x})||^2 = \sum_{i=1}^M \frac{m_i}{m} ||\nabla f_i(\mathbf{x}) - \nabla f(\mathbf{x})||^2 + \sum_{i=1}^M \frac{m_i}{m} \frac{1}{m_i} \sum_{j \in \mathcal{V}_i} ||\nabla F_i(\mathbf{x}) - \nabla f_i(\mathbf{x})||^2$, implying the asymptotic relation $\mathcal{O}(\tilde{\epsilon}^2) = \mathcal{O}(\epsilon^2 + \sum_{i=1}^M \frac{m_i}{m} \epsilon_i^2)$.

## 4.1 HFL WITH FWP

Consider the problem described in Section 3, we have the following results for HFL with FWP:

**Theorem 1.** *Under Assumption 1-4, with FWP, let the learning rates be chosen such that $\eta \leq \frac{1}{10I_{max}L}$, $\eta_c\eta \leq \frac{1}{10GL}$, $\eta_g\eta_c\eta \leq \frac{1}{GL}$, and $40G^2\eta_c^2\eta^2L^2 + 100\eta^2L^2\sum_{i=1}^{M}\frac{m_i}{m}I_i^2 < \frac{1}{2}$, where $I_{max} = \max_i I_i$, then the sequence of outputs $\{\overline{\mathbf{x}}^t\}$ generated by Algorithm 1 satisfies*

$$\min_{t\in[T]}\mathbb{E}\left\|\nabla f(\overline{\mathbf{x}}^t)\right\|^2 \leq \frac{f_0 - f_*}{c\eta_g\eta_c\eta GT} + \frac{1}{c}(\Phi_1 + \Phi_2).$$

*where $c$ is a constant, $f^0 \triangleq f(\overline{\mathbf{x}}^0)$, $f^* \triangleq f(\overline{\mathbf{x}}^*)$, and*

$$\Phi_1 = 9G\eta_c^2\eta^2L^2\frac{M}{m}\sigma^2 + 8\eta^2L^2\sum_{i=1}^{M}\frac{m_i}{m}I_i\sigma^2$$

$$+ 40G^2\eta_c^2\eta^2L^2\epsilon^2 + 100\eta^2L^2\sum_{i=1}^{M}\frac{m_i}{m}I_i^2\epsilon^2$$

$$+ 75\eta^2L^2\sum_{i=1}^{M}\frac{m_i}{m}I_i^2\epsilon_i^2 \, , \quad \Phi_2 = \frac{L\eta_g\eta_c\eta}{2m}\sigma^2.$$

*Proof.* Please refer to Appendix C. Here the core technique we use is the mutual bounding of worker-cluster parameter MSEs (WC-MSE) $\|\mathbf{x}_j^{t,\tau,h} - \overline{\mathbf{x}}_i^{t,\tau}\|^2$ and cluster-master parameter MSEs (CM-MSE) $\|\overline{\mathbf{x}}_i^{t,\tau} - \overline{\mathbf{x}}^t\|^2$, as shown in our derived Lemma 3 and 4. This may seem counterintuitive at first, since we may expect that it is only CM-MSE being bounded by WC-MSE, but not the contrary. However, this mutual bounding is exactly the maximal knowledge on MSEs when no stronger assumption is available (e.g., bounded gradient or convexity assumption). We creatively leverage this to further derive Lemma 5, which analytically provides a universal bound of WC-MSE. Then, with Lemma 5, we achieve the final convergence bound for HFL and recover the desired weakening effect. □

*Remark* 1. The convergence bound in Theorem 1 contains two parts: a vanishing term $\frac{f_0 - f_*}{c\eta_g\eta_c\eta GT}$ that decreases as $T$ increases, and other constants that depend on the problem instance configuration rather than $T$. The decaying rate of the vanishing term matches that of typical SGD methods. The constant part can be further categorized into two components $\Phi_1$ and $\Phi_2$ (this manual partition is for better comparison with subsequent results from PWP case). $\Phi_1$ reveals how the master and cluster periods $G, I_i$ interact with SGD noise $\sigma^2$ and divergences $\epsilon^2, \epsilon_i^2$. $\Phi_2$ covers all the impact of master learning rate $\eta_g$ for the constant part, which only acts on $\sigma^2$.

*Remark* 2. Theorem 1 shows how local aggregation of HFL helps to overcome divergences. The overall divergences for HFL are $\mathcal{O}(\eta^2\eta_c^2G^2\epsilon^2 + \eta^2\sum_{i=1}^{M}\frac{m_i}{m}I_i^2(\epsilon^2 +$

$\epsilon_i^2))$, originating from the constant component $\Phi_1$. In contrast, the corresponding divergence part in standard FL is $\mathcal{O}(\eta^2G^2\tilde{\epsilon}^2) = \mathcal{O}(\eta^2G^2\epsilon^2 + \eta^2G^2\sum_{i=1}^{M}\frac{m_i}{m}\epsilon_i^2)$ [Yang et al., 2021]. We suppose $\eta_c = \mathcal{O}(1)$ for fair comparison, then we simplify the divergences of HFL in asymptotic sense as $\mathcal{O}(\eta^2G^2\epsilon^2 + \eta^2\sum_{i=1}^{M}\frac{m_i}{m}I_i^2\epsilon_i^2)$. This indicates that local aggregation of HFL can weaken the impacts of the worker-cluster part of the global divergence since $G \geq I_i, \forall i \in [M]$. The weakening effect here matches the iteration-level convergence results of HFL from Wang et al. [2022]. Besides, our three-sided learning rates algorithm provides more flexibility here, since only the worker learning rate $\eta$ interacts with the cluster-master divergence $\epsilon_i$. This may allow a decoupling of learning, and we can thus adjust $\eta_c$ and $\eta$ according to divergences for probably better convergence. For example, larger $\epsilon_i$ and smaller $\epsilon$ may prefer a smaller $\eta$ for stability as well as a larger $\eta_c$ for acceleration.

**Corollary 1.** *Let $\eta = \frac{1}{\sqrt{T}GL}$, $\eta_c = \mathcal{O}(1)$, and $\eta_g = \sqrt{Gm}$. The convergence rate of the HFL with FWP in Algorithm 1 is $\mathcal{O}(\frac{1}{\sqrt{mGT}} + \frac{1}{T})$.*

*Remark* 3. The HFL with FWP achieves a linear speedup $\mathcal{O}(\frac{1}{\sqrt{mGT}})$ with proper learning rate settings as shown in Corollary 1 as long as $T \geq mG$. This resembles the results of standard FL from Yang et al. [2021]. To provide some flexibility, we set $\eta_c = \mathcal{O}(1)$ as we discussed in Remark 2, without impairing the linear speedup property.

## 4.2 HFL WITH PWP

Next, we analyze the convergence behavior for HFL with PWP, for which we have following results:

**Theorem 2.** *Under Assumption 1-4, with PWP, let the learning rates be chosen such that $\eta \leq \frac{1}{10I_{max}L}$, $\eta_c\eta \leq \frac{1}{10GL}$, and $\eta_g\eta_c\eta \leq \frac{1}{GL}$, then the sequence of outputs $\{\overline{\mathbf{x}}^t\}$ generated by Algorithm 1 satisfies*

$$\min_{t\in[T]}\mathbb{E}\left\|\nabla f(\overline{\mathbf{x}}^t)\right\|^2 \leq \frac{f_0 - f_*}{c\eta_g\eta_c\eta GT} + \frac{1}{c}(\Phi_1 + \Phi_2 + \Phi_3),$$

*where $c$ is a constant, $f^0 \triangleq f(\overline{\mathbf{x}}^0)$, $f^* \triangleq f(\overline{\mathbf{x}}^*)$, and for both sampling strategies*

$$\Phi_1 = 9G\eta_c^2\eta^2L^2\frac{M}{m}\sigma^2 + 8\eta^2L^2\sum_{i=1}^{M}\frac{m_i}{m}I_i\sigma^2$$

$$+ 40G^2\eta_c^2\eta^2L^2\epsilon^2 + 100\eta^2L^2\sum_{i=1}^{M}\frac{m_i}{m}I_i^2\epsilon^2$$

$$+ 75\eta^2L^2\sum_{i=1}^{M}\frac{m_i}{m}I_i^2\epsilon_i^2 \, , \quad \Phi_2 = \frac{1}{2}L\eta_g\eta_c\eta\sum_{i=1}^{M}\frac{m_i^2}{m^2n_i}\sigma^2.$$

*For strategy 1 (with replacement), let learning rates additionally satisfy $40G^2\eta_c^2\eta^2L^2 + 100\eta^2L^2\sum_{i=1}^{M}\frac{m_i}{m}I_i^2 +$*

$10\eta_g\eta_c\eta L \sum_{i=1}^{M} \frac{m_i^2}{m^2 n_i} I_i < \frac{1}{2}$, *it then holds that*

$$\Phi_3 = \frac{3}{4}\eta_g\eta_c\eta L \sum_{i=1}^{M} \frac{m_i^2}{m^2 n_i}\sigma^2 + \frac{15}{2}\eta_g\eta_c\eta L \sum_{i=1}^{M} \frac{m_i^2}{m^2 n_i} I_i\epsilon_i^2$$

$$+ 10\eta_g\eta_c\eta L \sum_{i=1}^{M} \frac{m_i^2}{m^2 n_i} I_i\epsilon^2.$$

*For strategy 2 (without replacement), let learning rates additionally satisfy* $40G^2\eta_c^2\eta^2 L^2 + 100\eta^2 L^2 \sum_{i=1}^{M} \frac{m_i}{m} I_i^2 + 10\eta_g\eta_c\eta L \sum_{i=1}^{M} \frac{m_i^2 \alpha_i}{m^2 n_i} I_i < \frac{1}{2}$, *where* $\alpha_i = \frac{m_i - n_i}{m_i - 1}$, *it then holds that*

$$\Phi_3 = \frac{3}{4}\eta_g\eta_c\eta L \sum_{i=1}^{M} \frac{m_i^2 \alpha_i}{m^2 n_i}\sigma^2 + \frac{15}{2}\eta_g\eta_c\eta L \sum_{i=1}^{M} \frac{m_i^2 \alpha_i}{m^2 n_i} I_i\epsilon_i^2$$

$$+ 10\eta_g\eta_c\eta L \sum_{i=1}^{M} \frac{m_i^2 \alpha_i}{m^2 n_i} I_i\epsilon^2.$$

*Proof.* Please refer to Appendix D. Regarding PWP, the core technique we use here is uncertainty redirection. In HFL, as the worker sampling occurs at each cluster round, the resulting uncertainty within a single master round accumulates across both multiple clusters and multiple rounds of a certain cluster. We creatively derive Lemma 6 to transform the corresponding accumulated uncertainty term $A_2$ in Eq. 32. We first decompose the impact of PWP among different clusters (inter-cluster) and then further decompose among different cluster rounds (intra-cluster). All these are carefully conducted via equality substitution. Besides, WC-MSE and CM-MSE shall both have their partial versions (not just reused from full case). We thus thoroughly renew our deduction for the partial case, consolidate it with the bounds for the full case, and present the results in Lemma 3 and 5. This further induces an additional variance term $\Psi_i^{t,\tau}$. We decompose it w.r.t. sampling strategy (Eq. 46 and 56) and merge into some resulting terms of Lemma 6, finally deriving the compact convergence bound [1]. □

*Remark* 4. The convergence bound in Theorem 2 shows certain consistency with that of Theorem 1, as they share the same constant component $\Phi_1$. The uncertainty introduced by PWP contributes to amplifying constant $\Phi_2$, and incurring an additional $\Phi_3$. Especially, for sampling strategy 2 with sampling size $n_i = m_i, \forall i \in [M]$, Theorem 2 would as expected, recover exactly the same convergence bound with the FWP case.

**Corollary 2** (Minimum Convergence Rate). *Let* $\eta = \frac{1}{\sqrt{TGL}}$, $\eta_c = \mathcal{O}(1)$, *and* $\eta_g = \sqrt{Gm\kappa_{min}}$, *where* $\kappa_{min} = \min_i \frac{n_i}{m_i}$. *The minimum convergence rate of the HFL with PWP in*

Algorithm 1 for both sampling strategies is $\mathcal{O}(\frac{I_{max}}{\sqrt{m\kappa_{min}GT}} + \frac{1}{T})$.

*Remark* 5. HFL with PWP can at least achieve a linear speedup $\mathcal{O}(\frac{I_{max}}{\sqrt{m\kappa_{min}GT}})$ with proper learning rate settings as shown in Corollary 2. The minimum convergence rate of the HFL system is bottlenecked by the minimal sampling rate $\kappa_{min}$ and the maximal cluster period $I_{max}$.

Note that the minimum convergence rate mentioned above is just the loosest estimation. In the following, we will show a more typical case.

**Corollary 3.** *Suppose* $\frac{n_i}{m_i} = \frac{n}{m}, I_i = I, \forall i \in [M]$, *and let* $\eta = \frac{1}{\sqrt{TGL}}$, $\eta_c = \mathcal{O}(1)$, *and* $\eta_g = \sqrt{Gn}$. *The convergence rate of the hierarchical FL with PWP in Algorithm 1 for both sampling strategies is* $\mathcal{O}(\frac{I}{\sqrt{nGT}} + \frac{1}{T})$.

*Remark* 6. When all clusters have the same (or close) sampling rates and round period, a linear speedup $\mathcal{O}(\frac{I}{\sqrt{nGT}})$ can be guaranteed by setting the learning rates as shown in Corollary 3. Note that the convergence rate here has a smaller first term than that of standard FL with PWP, which is $\mathcal{O}(\frac{\sqrt{G}}{\sqrt{nT}} + \frac{1}{T})$ (Corollary 2 from Yang et al. [2021]). This indicates the additional benefit of PWP on HFL. However, this homogeneous setting[2] of HFL may not necessarily lead to the optimal convergence rate, which depends on the specific worker-cluster divergence $\epsilon_i^2$. We will subsequently discuss this further.

*Remark* 7. The convergence rate bound for HFL with PWP has the same structure (in order sense) as the full case, but with a larger variance. This is consistent with the results for standard FL [Yang et al., 2021]. Uniform sampling (with/without replacement) yields a good approximation of the entire intra-group worker distribution in expectation, thereby reducing the risk of distribution deviation incurred by PWP.

## 4.3 OVERCOME DIVERGENCE

We now exclusively elaborate how the weakening effect for overcoming divergences in HFL, as shown in FWP case (Remark 2), can be enhanced with PWP. Specifically, we focus on the additional divergences resulting from PWP (covered by component $\Phi_3$) in HFL, denoted as $\Theta_H$. For strategy 1, we have

$$\Theta_H = \mathcal{O}(\eta_g\eta_c\eta \sum_{i=1}^{M} \frac{m_i^2}{m^2 n_i} I_i(\epsilon_i^2 + \epsilon^2)),$$

---

[1] In FL with PWP, the analysis is much explicit due to the absence of intermediate cluster aggregation (thus there is no accumulation effect of $A_2$ term or additional randomness left in MSE term).

[2] Similar setting is also considered in Qu et al. [2022] for multi-server federated learning (but non-hierarchical), where they refer to as unbiased PWP. However, here we do not exclusively emphasize this, since it could be inherently covered by our theoretical results.

Table 2: Communication time (s) and iterations ($\times 10^4$) to achieve target test accuracy with 20% workers participating.

| | | Standard FL ($P$) | | | HFL ($G, I$) | | |
|---|---|---|---|---|---|---|---|
| MNIST | Setting | 10 | 50 | 100 | 50, 10 | 100, 10 | 100, 50 |
| | Communication Time | 15.95 | 11.48 | 9.91 | 1.5 | 1.54 | 1.09 |
| | Iterations ($\times 10^4$) | 1.46 | 5.27 | 9.09 | 1.38 | 1.4 | 5.15 |
| FEMNIST | Setting | 20 | 100 | 200 | 100, 20 | 200, 20 | 200, 100 |
| | Communication Time | 8.92 | 3.03 | 2.78 | 0.93 | 1.01 | 0.34 |
| | Iterations ($\times 10^4$) | 0.73 | 1.24 | 2.24 | 0.77 | 0.82 | 1.3 |
| CIFAR-10 | Setting | 10 | 50 | 250 | 50, 10 | 250, 10 | 250, 50 |
| | Communication Time | 399.77 | 113.03 | 54.27 | 49.29 | 50.57 | 18.79 |
| | Iterations ($\times 10^4$) | 1.24 | 1.76 | 4.23 | 1.54 | 1.58 | 2.93 |

and for strategy 2, we have

$$\Theta_H = \mathcal{O}(\eta_g \eta_c \eta \sum_{i=1}^{M} \frac{m_i^2 \alpha_i}{m^2 n_i} I_i(\epsilon_i^2 + \epsilon^2)).$$

For standard FL, Theorem 2 of Yang et al. [2021] indicates that the additional divergence part resulting from partial worker participation is, for sampling strategy 1, we have

$$\Theta_S = \mathcal{O}(\frac{1}{n}\eta_g \eta G \tilde{\epsilon}^2), \tag{2}$$

and for sampling strategy 2, we have

$$\Theta_S = \mathcal{O}(\frac{\alpha}{n}\eta_g \eta G \tilde{\epsilon}^2), \tag{3}$$

where $\alpha = \frac{m-n}{m-1}$. Here we merge their quadratic term for better analysis.

With PWP, the weakening effect is observed for both the worker-cluster and cluster-master part. Taking strategy 1 as example, when $\frac{n_i}{m_i} = \frac{n}{m}, I_i = I, \forall i \in [M]$ and $\eta_c = \mathcal{O}(1)$, it always holds in the asymptotic sense that

$$\Theta_H = \mathcal{O}(\eta_g \eta \sum_{i=1}^{M} \frac{m_i}{mn} I(\epsilon_i^2 + \epsilon^2))$$
$$= \mathcal{O}(\frac{1}{n}\eta_g \eta I \tilde{\epsilon}^2) < \mathcal{O}(\frac{1}{n}\eta_g \eta G \tilde{\epsilon}^2) = \Theta_S.$$

The same holds for strategy 2. Compared to standard FL, this suggests that setting the same (or close) sampling rates for all clusters can consistently reduce additional divergences. As we discussed in Remark 2, this weakening effect yet is only observed on worker-cluster divergences in HFL with FWP. In fact, the bound $\mathcal{O}(\frac{1}{n}\eta_g \eta I \tilde{\epsilon}^2)$ indicates that this weakening effect can always restrict the global divergences $\tilde{\epsilon}^2$ to only being intensified by $I$ rather than $G$, regardless of the specific grouping setting (i.e., grouping-agnostic). Therefore, PWP can probably guarantee the performance of HFL with $G, I$ to be close to that of standard FL with aggregation period $I$. We empirically verify this superiority of HFL with PWP from our experiments in Section 5.

Though $\frac{n_i}{m_i} = \frac{n}{m}, \forall i \in [M]$ may not be the optimal solution, it can be used as a priori to serve as a sufficient condition for

reducing additional divergences. Note that we can flexibly set $\frac{n_i}{m_i}$ to match $I_i$ and $\epsilon_i$ accordingly for a better weakening effect. For instance, in practice, some clusters may inherently have larger inner divergence. In this case, enabling a larger sampling size and a smaller round period (if possible) could probably ensure a more effective convergence.

## 5 NUMERICAL EXPERIMENTS

We conduct extensive experiments to validate our theoretical results. We defer some results to Appendix E. All reported results are averaged over five random realizations.

Table 3: Per-round communication time between worker and cluster.

| Model | CNN (MNIST) | CNN (FEMNIST) | PreAct ResNet-18 |
|---|---|---|---|
| RTT (ms) | 1.09±0.17 | 2.45±0.56 | 32.11±6.32 |

### 5.1 DATASET

In our experiments, we choose three real datasets: MNIST, FEMNIST, and CIFAR-10. We partition the three datasets in non-i.i.d. manner, with details as follows

- **MNIST**. The MNIST dataset LeCun et al. [1998] consists of images of handwritten digits 0-9, with 60,000 training samples and 10,000 test samples. We distribute the training data to $m = 100$ workers uniformly. We restrict each worker to have training samples of no more than 2 class of digits, to provide certain data heterogeneity.

- **FEMNIST**. FEMNIST is a federated version of the EMNIST dataset proposed by LEAF Caldas et al. [2018]. We follow the non-i.i.d. preprocessing protocol of Wang et al. [2022], where the training set consists of 34,659 samples distributed to $m = 156$ workers and the test set consists of 4,973 samples.

- **CIFAR-10**. The CIFAR-10 dataset Krizhevsky et al. [2009] consists of $32 \times 32$ color images in 10 classes,

with 50,000 training samples and 10,000 test samples. Like we do in MNIST, we distribute the training data to $m = 100$ workers uniformly and restrict each worker to have training samples of no more than 4 class, to provide certain data heterogeneity.

For the group non-i.i.d. setting, we restrict each group to have worker training samples of no more than 4 classes on MNIST, while no more than 6 classes on CIFAR-10.

## 5.2 IMPLEMENTATION DETAILS

We use Python 3.7 with PyTorch 1.8.1 to implement all our models and HFL algorithm[3]. We set the learning rates $\eta = 0.01$, $\eta_c = 1$ and $\eta_g = 1$. The local SGD mini-batch of each worker is set to 20.

We use CNNs for both MNIST and FEMNIST datasets. Specifically, for MNIST, we use CNN model composed of two convolutional layers and two fully connected layers. For FEMNIST, we use the same architecture as Wang et al. [2022]. For CIFAR-10, we use PreAct ResNet-18 He et al. [2016].

For the communication time, we follow the emulation of Wang et al. [2022]. Specifically, we measure the round-trip time (RTT) of transmitting the model between an end device (worker) and a nearby server (cluster). Due to resource limitation, we simply assume the worker-master RTT is ten times as the worker-cluster one (which basically matches Wang et al. [2022], Liu et al. [2020]). The estimated worker-cluster RTT is presented in Table 3.

## 5.3 COMMUNICATION OVERHEAD

Table 2 presents a comparison between standard FL and HFL on three datasets, all with PWP. $P$ stands for aggregation period of standard FL. The target test accuracy are $95\%$ for MNIST, $80\%$ for FEMNIST, and $85\%$ for CIFAR-10. By default, we uniformly partition workers into 4 groups.

We observe that HFL can benefit the training process in terms of reducing communication overhead. HFL basically shows a similar convergence performance to its standard FL counterpart with $P = I$, even when $G$ is large. In particular, on FEMNIST, the total number of iterations ($\times 10^4$) of standard FL for $P = 20$ is 0.73, only slightly less than 0.77 and 0.82, from HFL with $G = 100, I = 20$ and $G = 200, I = 20$, respectively. Note that on MNIST, HFL is even better. On the other hand, HFL requires much less communication time (about only one-tenth) to achieve certain target test accuracy, due to the ultra-low communication latency granted by parallel local aggregations.

---

[3]Our code is available at `https://github.com/cardistryj/HFL`.

## 5.4 WEAKENING EFFECT

Fig. 1 shows the convergence curves in terms of test accuracy on MNIST. Group i.i.d. stands for small cluster master divergence grouping, i.e., $\epsilon^2 \approx 0$, while group non-i.i.d. for large $\epsilon^2$ setting. For PWP, we always keep $20\%$ sampling rate.

We first focus on FWP. From Fig. 1(c), we observe that the convergence performance of HFL with $G$ and $I$ is between that of standard FL with $P = I$ and with $P = G$ (also referred to as "sandwitch" behavior from Wang et al. [2022]). This matches the weakening effect of local aggregation on worker-cluster divergences as we discussed in Remark 2. Note that the performance gap between HFL and its standard FL counterpart with $P = I$ originates from the amplified impact of $\epsilon^2$ by $G$ as shown in our Theorem 1 ($\sigma^2$ is also negligible compared to non-i.i.d. divergences). To verify this, we refer to the corresponding curves in Fig. 1(a), observing almost the same convergence trends (e.g., $G = 50, I = 10$ and $G = 100, I = 10$ versus $P = 10$). Essentially, it is the group i.i.d. setting with $\epsilon^2 \approx 0$ that makes the impact of $G$ trivial and eliminates the aforementioned performance gap.

With insights above, we next check on PWP. For group i.i.d. setting in Fig. 1(b), the curve patterns show high consistency with those in Fig. 1(a), where there is also no performance gap between HFL and its standard FL counterpart with $P = I$. Still, PWP introduces additional randomness, resulting in zigzagging curves and slower convergence. For group non-i.i.d. setting, the results are more significant. The curve patterns in Fig. 1(d), instead resembles those in Fig. 1(b) (while unlike those in Fig. 1(c)), i.e., without noticeable performance gap. This exactly matches the enhanced weakening effect with PWP on global divergences (especially the cluster-master part) as we discussed in the last part of Section 4.2. Intuitively, PWP pushes the convergence behavior of HFL with $G, I$ to the optimal upper boundary of the "sandwitch" (i.e., standard FL with $P = I$). Therefore, it could always be a beneficial choice for HFL to enable PWP.

## 5.5 COMPARISON WITH OTHER METHODS

We conduct comparison experiments on MNIST dataset to justify the effectiveness of our proposed three-sided learning rates HFL algorithm. We use the group non-i.i.d. setting with 4 groups and $20\%$ workers participating. The round periods are set to $G = 100, I = 10$.

We mainly consider three works as follows

- **Hier-Local-QSGD** [Liu et al., 2022a]: An HFL optimization algorithm with model quantization to reduce communication overhead. We conduct quantization by converting the weights from fp32 into int8. RTT for the quantized model is $0.37 \pm 0.1$ms.

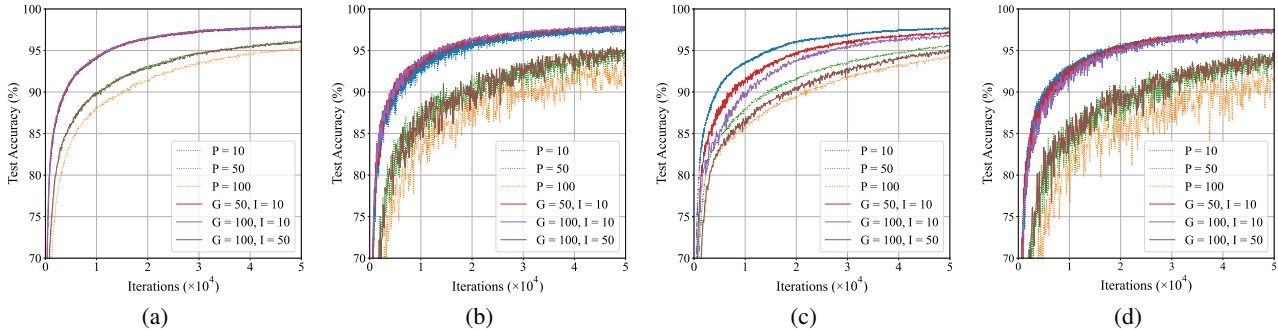

Figure 1: Test Accuracy w.r.t. iterations on MNIST. (a) Group i.i.d. with full participation; (b) Group i.i.d. with partial participation; (c) Group non-i.i.d. with full participation; (d) Group non-i.i.d. with partial participation.

Table 4: Comparison among different methods to achieve target accuracy on MNIST.

|  | Iterations ($\times 10^4$) | Communication Time |
|---|---|---|
| Hier-Local-QSGD | 3.57 | 1.32 |
| HierMo | 0.24 | 0.47 |
| MLL-SGD (fully-connected) | 2.23 | 2.43 |
| MLL-SGD (ring) | 3.1 | 3.38 |
| HierFedAvg | 2.22 | 2.42 |
| Ours ($\eta_c = 1, \eta_g = 2$) | 1.95 | 2.13 |
| Ours ($\eta_c = 1, \eta_g = 3$) | 1.68 | 1.83 |
| Ours ($\eta_c = 3, \eta_g = 1$) | 0.29 | 0.32 |
| Ours ($\eta_c = 5, \eta_g = 1$) | **0.22** | **0.24** |
| Ours ($\eta_c = 3, \eta_g = 2$) | 0.57 | 0.62 |

Table 5: Impact of cluster period and sampling number to achieve target test accuracy on MNIST.

| Cluster periods ($I_1$, $I_2$) | 10, 200 | 20, 100 | 50, 50 | 100, 20 | 200, 10 |
|---|---|---|---|---|---|
| Communication Time | 1.18 | 0.68 | 0.42 | 0.64 | 0.93 |
| Master Rounds | 46 | 47 | 47 | 44 | 37 |
| Sampling Number ($n_1$, $n_2$) | 1, 19 | 6, 14 | 10, 10 | 14, 6 | 19, 1 |
| Communication Time | 0.46 | 0.42 | 0.46 | 0.56 | 1.40 |
| Master Rounds | 52 | 47 | 51 | 62 | 157 |

- **HierMo** [Yang et al., 2023]: An HFL optimization algorithm with momentum update to accelerate the convergence of HFL. We use the provided setting in the original paper. Due to the extra transmission of momentum, RTT for HierMo is 1.96±0.3ms.
- **MLL-SGD** [Castiglia et al., 2020]: A partially decentralized FL algorithm, where it is still a two-level architecture while clusters are organized as a peer-to-peer network. We consider both fully-connected topology and ring topology for the cluster network.

We also adjust cluster learning rate $\eta_c$ and master learning rate $\eta_g$ of our algorithm. The configuration with $\eta_c = 1$ and $\eta_g = 1$ can also be considered as a natural generalization of FedAvg in HFL (referred to as HierFedAvg). We maintain the same worker learning rate $\eta = 0.01$ for all the methods mentioned above.

Table 4 presents the performance among different methods.

We highlight the best results in bold style, while the second best with underline.

We can observe the effectiveness of our three-sided learning rates. When tuning $\eta_c$ and $\eta_g$, there are varying degrees in acceleration on convergence. The speedup effect of $\eta_c$ is particularly pronounced. Simply setting $\eta_c$ to 3 or 5 reduces the iterations required to achieve target accuracy from the original $2.22 \times 10^4$ of HierFedAvg to less than 3000, indicating a nearly tenfold acceleration. The best hyperparameter combinations require a more refined tuning and searching process. However, we can still observe the significant potential of adjusting $\eta_c$ and $\eta_g$ to facilitate convergence at no additional information or communication cost.

HierMo also performs well, achieving target accuracy with the second fewest iterations. However, this acceleration comes at the cost of more communication overhead due to the momentum update. In contrast, Hier-Local-QSGD

can directly mitigate the communication overhead by quantizing and compressing model weights. Nevertheless, the introduction of quantization leads to information loss, consequently impeding the convergence performance. For MLL-SGD algorithm, we observe that its fully-connected variant is essentially equivalent to the HFL architecture. The only difference is that the aggregation among clusters is achieved through peer-to-peer communication rather than through the coordination of a master node. This can be verified by the very close performance between MLL-SGD (fully-connected) and HierFedAvg. MLL-SGD (ring) exhibits relatively slower convergence rate. This is because the model aggregation among clusters cannot always be fully synchronized.

## 5.6  HETEROGENEOUS GROUPS

Table 5 shows the impact of cluster period and sampling ratio on MNIST. Here we use two groups with 30 and 70 workers, respectively. Group 1 is i.i.d. partitioned among workers (i.e., small $\epsilon_1^2$) while group 2 is non-i.i.d. partitioned (i.e., large $\epsilon_2^2$). The default setting is $G = 200$, $(I_1, I_2) = (50, 50)$, and $(n_1, n_2) = (6, 14)$, namely, a homogeneous clusters setting in Corollary 3. We always keep an invariant $n = n_1 + n_2 = 20$.

Though a small $I_2 = 10$ could save for about 10 master rounds, this is at cost of incurring more communication time instead. Note that impact of cluster period is not as that explicit as sampling ratios. In the extreme case $n_2 = 1$, the number of master rounds increases to about three times. Hence, choosing proper configurations is important to achieve a fair convergence. Still, the default homogeneous setting could sufficiently serve as a decent solution.

## 6  DISSCUSSION

**Flexible Hierarchical Structure:** For edge-based FL, clients within the communication range of the server collaborate to train a machine learning model. Generally, the location and communication range of edge servers (such as base stations) are fixed. Treating the clients within the communication range of an edge server as one cluster is a natural choice. Therefore, in this paper, we employ given clusters (i.e., assuming an arbitrary grouping) which is practical in real-world scenarios. Regarding sophisticated hierarchical structure design (such as edge server deployment, grouping strategies), it is important and worth researching.

On the other hand, our theoretical framework allows certain flexibility and individualization for HFL. This includes the master round $G$ for the whole learning system, different worker number $m_i$, sampling number $n_i$, and cluster period $I_i$ for each cluster. Still, different cluster may possess inherent characteristics, such as data heterogeneity. We dis-

cuss this in subsection 4.3, where each cluster can adjust its worker sampling ratio and round period accordingly to deal with its inner data heterogeneity. We also verify this with experiments in subsection 5.6.

**More Levels:** The cloud-edge-end architecture is prevalent in edge computing, constituting a two-level architecture. Therefore, our work delves into the convergence performance of two-level HFL. For HFL with more than 2 levels, we assume that there are $\mathcal{L}$ levels in total. The global server is at the uppermost level $l = 1$. Each upper-level server at level $l = 1, \ldots, \mathcal{L} - 1$ connects to $M_l$ lower-level servers at level $l + 1$. At the lowest level $l = \mathcal{L}$, each edge server serves $M_{\mathcal{L}}$ clients. Intuitively, we can straightforwardly extend our theoretical results. The weakening effects will still apply to each connected level in multi-level HFL with PWP, aiding in reducing data divergences. The global divergences $\tilde{\epsilon}^2$ are expected to only be intensified by the lowest level aggregation period $I_{\mathcal{L}}$ rather than global period $G$.

**Communication Time:** If there is minimal disparity in worker-master RTT and worker-cluster RTT, the communication costs of HFL and standard FL [Yang et al., 2021] will be nearly identical. On the other hand, a critical factor driving the widespread adoption of HFL is its low latency. By distributing computational tasks across cloud servers, edge servers, and clients, data processing can occur closer to its source, reducing the latency associated with transmitting data back and forth to distant cloud servers.

## 7  CONCLUSION

In this work, we study the convergence behavior of HFL. We newly derive a general convergence bound for HFL that covers both full and PWP with non-i.i.d. data, non-convex objective function and SGD. Based on the convergence analysis, we develop a three-sided learning rates algorithm to mitigate data divergences issue and realize better convergence performance. Our theoretical results provide key insights of why PWP of HFL is beneficial in significantly reducing the data divergences compared to standard FL. Besides, we provide a degree of individualization for each cluster in HFL, indicating that adjusting the worker sampling ratio and round period to match inner divergence can potentially improve the convergence behavior. We conduct extensive experiments on real world datasets to verify our theoretical results.

### Acknowledgements

This work was supported in part by the National Natural Science Foundation of China (NSFC) under Grant 62306077, the National Key Research and Development Program of China under Grant 2023YFC3305304, and Shanghai Sailing Program under Grant 23YF1402600.

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

# A   NOTATION TABLE

Table 6: Key notations for three-sided learning rates HFL algorithm.

| | |
|---|---|
| $\eta$ | worker learning rate |
| $\eta_c$ | cluster learning rate |
| $\eta_g$ | master learning rate |
| $T$ | number of total master rounds |
| $G$ | master aggregation period |
| $I_i$ | cluster $i$ aggregation period |
| $\omega_i$ | number of cluster rounds for $i$ in a master round |
| $t$ | the index of master round, $0 \le t < T$ |
| $\tau$ | the index of cluster round, $0 \le \tau < \omega_i$ |
| $h$ | the index of worker local iteration, $0 \le h < I_i$ |
| $\mathcal{V}_i$ | set of workers in cluster $i$ with size $m_i$ |
| $\mathcal{S}_i^{t,\tau}$ | set of workers in cluster $i$ sampled in master round $t$ and cluster round $\tau$ with size $n_i$ |
| $\bar{\mathbf{x}}^t$ | master aggregated parameters before master round $t$ |
| $\bar{\mathbf{x}}_i^{t,\tau}$ | aggregated parameters on cluster $i$ before master round $t$ and cluster round $\tau$ |
| $\mathbf{x}_j^{t,\tau,h}$ | local model parameters on worker $j$ at update step $(t,\tau,h)$, where the total number of iterations is $Gt + I_i\tau + h$ |

# B   PRELIMINARY OF PROOF

**Lemma 1** (Unbiased Sampling)**.** *For both sampling strategies 1 and 2, the estimator is unbiased in both cluster and master round, i.e.,*

$$\mathbb{E}[\Delta_i^{t,\tau}] = \bar{\Delta}_i^{t,\tau}, \forall i \in [M], \text{ and } \mathbb{E}[\Delta^t] = \bar{\Delta}^t,$$

*where the expectation is taken over the randomness introduced by sampling workers.*

*Proof.* Let $\mathcal{S}_i^{t,\tau} = \{l_{i,1}, l_{i,2}, \ldots, l_{i,n_i}\}$ with size $n_i$. For both sampling strategies 1 and 2, each sampling distribution is identical. Therefore, for each cluster $i \in [M]$, we have

$$\mathbb{E}[\Delta_i^{t,\tau}] = \frac{1}{n_i}\mathbb{E}\Big[\sum_{l_{i,j} \in \mathcal{S}_i^{t,\tau}} \tilde{\Delta}_{l_{i,j}}^{t,\tau}\Big] = \mathbb{E}[\tilde{\Delta}_{l_{i,1}}^{t,\tau}] = \frac{1}{m_i}\sum_{j \in \mathcal{V}_i} \tilde{\Delta}_j^{t,\tau} = \bar{\Delta}_i^{t,\tau}. \tag{4}$$

And we have for the master

$$\mathbb{E}[\Delta^t] = \sum_{i=1}^{M} \frac{m_i}{m} \sum_{\tau=0}^{\omega_i-1} \eta_c \mathbb{E}[\Delta_i^{t,\tau}] = \sum_{i=1}^{M} \frac{m_i}{m} \sum_{\tau=0}^{\omega_i-1} \eta_c \bar{\Delta}_i^{t,\tau} = \bar{\Delta}^t. \tag{5}$$

This completes the proof.                                                                                          □

Note that this unbiased sampling property also inherently applies to the full gradient $\nabla F_j(\mathbf{x}_j^{t,\tau,h})$. This is guaranteed by taking additional expectation over the stochastic gradient, which is independent of the worker sampling.

**Lemma 2** (Lemma 7 from Reddi et al. [2020])**.** *For independent, mean 0 random variables $z_1, ..., z_r$ we have*

$$\mathbb{E}\left[\|z_1 + ... + z_r\|^2\right] = \mathbb{E}\left[\|z_1\|^2 + ... + \|z_r\|^2\right]. \tag{6}$$

# C   PROOF OF THEOREM 1

*Proof.* We start with bounding for the full participation case, where $\Delta^t$ exactly equals to $\bar{\Delta}^t$. Taking expectation over the randomness of the master round $t$, we have

$$\mathbb{E}_t[f(\bar{\mathbf{x}}^{t+1})] = \mathbb{E}_t f\left(\mathbf{x}^t - \eta_g \Delta^t\right)$$

$$\overset{(a)}{\leq} \mathbb{E}_t f(\overline{\mathbf{x}}^t) - \mathbb{E}_t\left\langle \nabla f(\overline{\mathbf{x}}^t), \eta_g \Delta^t \right\rangle + \frac{L}{2} \mathbb{E}_t \left\| \eta_g \Delta^t \right\|^2$$

$$= f(\overline{\mathbf{x}}^t) - \eta_g \mathbb{E}_t\left\langle \nabla f(\overline{\mathbf{x}}^t), \Delta^t - \eta_c \eta G \nabla f(\overline{\mathbf{x}}^t) + \eta_c \eta G \nabla f(\overline{\mathbf{x}}^t) \right\rangle + \eta_g^2 \frac{L}{2} \mathbb{E}_t \left\| \Delta^t \right\|^2$$

$$= f(\overline{\mathbf{x}}^t) - \eta_g \eta_c \eta G \left\| \nabla f(\overline{\mathbf{x}}^t) \right\|^2 - \eta_g \underbrace{\mathbb{E}_t\left\langle \nabla f(\overline{\mathbf{x}}^t), \Delta^t - \eta_c \eta G \nabla f(\overline{\mathbf{x}}^t) \right\rangle}_{A_1} + \eta_g^2 \frac{L}{2} \underbrace{\mathbb{E}_t \left\| \Delta^t \right\|^2}_{A_2}, \tag{7}$$

where $(a)$ is a proposition of Lipschitz smooth.

The inner product term $A_1$ follows

$$A_1 = \mathbb{E}_t\left\langle \nabla f(\overline{\mathbf{x}}^t), \sum_{i=1}^{M} \frac{m_i}{m} \sum_{\tau=0}^{\omega_i-1} \eta_c \sum_{h=0}^{I_i-1} \frac{1}{m_i} \sum_{j\in\mathcal{V}_i} \eta \mathbf{g}_j^{t,\tau,h} - \eta_c \eta G \nabla f(\overline{\mathbf{x}}^t) \right\rangle$$

$$= \mathbb{E}_t\left\langle \nabla f(\overline{\mathbf{x}}^t), \sum_{i=1}^{M} \frac{m_i}{m} \sum_{\tau=0}^{\omega_i-1} \eta_c \sum_{h=0}^{I_i-1} \frac{1}{m_i} \sum_{j\in\mathcal{V}_i} \eta \nabla F_j(\mathbf{x}_j^{t,\tau,h}) - \eta_c \eta G \nabla f(\overline{\mathbf{x}}^t) \right\rangle$$

$$= \mathbb{E}_t\left\langle \sqrt{\eta_c \eta G} \nabla f(\overline{\mathbf{x}}^t), \frac{\sqrt{\eta_c \eta}}{\sqrt{G}} \sum_{i=1}^{M} \frac{m_i}{m} \sum_{\tau=0}^{\omega_i-1} \sum_{h=0}^{I_i-1} \frac{1}{m_i} \sum_{j\in\mathcal{V}_i} \left( \nabla F_j(\mathbf{x}_j^{t,\tau,h}) - \nabla F_j(\overline{\mathbf{x}}^t) \right) \right\rangle$$

$$\overset{(a)}{=} -\frac{\eta_c \eta G}{2} \left\| \nabla f(\overline{\mathbf{x}}^t) \right\|^2 - \frac{\eta_c \eta}{2G} \mathbb{E}_t \left\| \sum_{i=1}^{M} \frac{m_i}{m} \sum_{\tau=0}^{\omega_i-1} \sum_{h=0}^{I_i-1} \frac{1}{m_i} \sum_{j\in\mathcal{V}_i} \left( \nabla F_j(\mathbf{x}_j^{t,\tau,h}) - \nabla F_j(\overline{\mathbf{x}}^t) \right) \right\|^2$$

$$+ \frac{\eta_c \eta}{2G} \mathbb{E}_t \left\| \sum_{i=1}^{M} \frac{m_i}{m} \sum_{\tau=0}^{\omega_i-1} \sum_{h=0}^{I_i-1} \frac{1}{m_i} \sum_{j\in\mathcal{V}_i} \nabla F_j(\mathbf{x}_j^{t,\tau,h}) \right\|^2, \tag{8}$$

where $(a)$ is due to the fact $<x, y> = \frac{1}{2}(\|x+y\|^2 - \|x\|^2 - \|y\|^2)$.

For the second term of Eq. 8, we have

$$\mathbb{E}_t \left\| \sum_{i=1}^{M} \frac{m_i}{m} \sum_{\tau=0}^{\omega_i-1} \sum_{h=0}^{I_i-1} \frac{1}{m_i} \sum_{j\in\mathcal{V}_i} \left( \nabla F_j(\mathbf{x}_j^{t,\tau,h}) - \nabla F_j(\overline{\mathbf{x}}^t) \right) \right\|^2$$

$$\overset{(a)}{\leq} \sum_{i=1}^{M} \frac{m_i}{m} G \sum_{\tau=0}^{\omega_i-1} \sum_{h=0}^{I_i-1} \frac{1}{m_i} \sum_{j\in\mathcal{V}_i} \mathbb{E}_t \left\| \nabla F_j(\mathbf{x}_j^{t,\tau,h}) - \nabla F_j(\overline{\mathbf{x}}^t) \right\|^2$$

$$= G \frac{1}{m} \sum_{i=1}^{M} \sum_{\tau=0}^{\omega_i-1} \sum_{h=0}^{I_i-1} \sum_{j\in\mathcal{V}_i} \mathbb{E}_t \left\| \nabla F_j(\mathbf{x}_j^{t,\tau,h}) - \nabla F_j(\overline{\mathbf{x}}_i^{t,\tau}) + \nabla F_j(\overline{\mathbf{x}}_i^{t,\tau}) - \nabla F_j(\overline{\mathbf{x}}^t) \right\|^2$$

$$\overset{(b)}{\leq} 2L^2 G \frac{1}{m} \sum_{i=1}^{M} \sum_{\tau=0}^{\omega_i-1} \sum_{h=0}^{I_i-1} \sum_{j\in\mathcal{V}_i} \mathbb{E}_t \left\| \mathbf{x}_j^{t,\tau,h} - \overline{\mathbf{x}}_i^{t,\tau} \right\|^2 + 2L^2 G \sum_{i=1}^{M} \sum_{\tau=0}^{\omega_i-1} I_i \frac{m_i}{m} \mathbb{E}_t \left\| \overline{\mathbf{x}}_i^{t,\tau} - \overline{\mathbf{x}}^t \right\|^2, \tag{9}$$

where $(a)$ is a proposition of Jensen's Inequality. $(b)$ is achieved by first unrolling with an extension of Jensen inequality as $\| \sum_i^k \mathbf{x}_i \|^2 \leq k \sum_i^k \|\mathbf{x}_i\|^2$ and then applying Assumption 1.

In Eq. 9, the first term represents the overall worker-cluster parameter MSE while the second one represents the overall cluster-master parameter MSE in each round.

## C.1 BOUNDING CLUSTER-MASTER PARAMETER MSE

We prove a lemma that bounds this cluster-master parameter MSE term.

**Lemma 3** (Cluster-master Parameter MSE). *For any local learning rate $\eta$ and cluster learning rate $\eta_c$ satisfying $\eta_c \eta \leq \frac{1}{8LG}$, we can bound the overall cluster-master parameter MSE for a cluster $i$ in a certain master round $t$ regarding the worker-*

*cluster parameter MSE as, with full worker participation,*

$$\sum_{\tau=0}^{\omega_i-1} \mathbb{E}_t \left\| \bar{\mathbf{x}}_i^{t,\tau} - \bar{\mathbf{x}}^t \right\|^2 \le 5G\eta_c^2\eta^2 \frac{\omega_i}{m_i}\sigma^2 + 40\omega_i G^2\eta_c^2\eta^2\epsilon^2$$

$$+ 40\omega_i G^2\eta_c^2\eta^2 \left\| \nabla f(\bar{\mathbf{x}}^t) \right\|^2 + 24G\eta_c^2\eta^2 L^2\omega_i \sum_{\tau=0}^{\omega_i-1} \Omega_i^{t,\tau}, \tag{10}$$

*where $\Omega_i^{t,\tau} = \frac{1}{m_i}\sum_{j\in\mathcal{V}_i}\sum_{h=0}^{I_i}\mathbb{E}\|\mathbf{x}_j^{t,\tau,h} - \bar{\mathbf{x}}_i^{t,\tau}\|^2$.*

*With partial worker participation for both sampling strategies, we have*

$$\sum_{\tau=0}^{\omega_i-1} \mathbb{E}_t \left\| \bar{\mathbf{x}}_i^{t,\tau} - \bar{\mathbf{x}}^t \right\|^2 \le 5G\eta_c^2\eta^2 \frac{\omega_i}{n_i}\sigma^2 + 40\omega_i G^2\eta_c^2\eta^2\epsilon^2$$

$$+ 40\omega_i G^2\eta_c^2\eta^2 \left\| \nabla f(\bar{\mathbf{x}}^t) \right\|^2 + 24G\eta_c^2\eta^2 L^2\omega_i \sum_{\tau=0}^{\omega_i-1} \Omega_i^{t,\tau} + 3\eta_c^2\eta^2\omega_i \sum_{\tau=0}^{\omega_i-1} \Psi_i^{t,\tau}, \tag{11}$$

*where $\Psi_i^{t,\tau} = \mathbb{E}\|\frac{1}{n_i}\sum_{j\in\mathcal{S}_i^{t,\tau}}\sum_{h=0}^{I_i}\nabla F_j(\mathbf{x}_j^{t,\tau,h}) - \frac{1}{m_i}\sum_{j\in\mathcal{V}_i}\sum_{h=0}^{I_i}\nabla F_j(\mathbf{x}_j^{t,\tau,h})\|^2$.*

*Proof.* We elaborate the proof for the partial worker participation, and the result naturally generalize to the full case. For any round $\tau$ of certain cluster $i$, we have

$$\mathbb{E}_t \left\| \bar{\mathbf{x}}_i^{t,\tau} - \bar{\mathbf{x}}^t \right\|^2 = \mathbb{E} \left\| \bar{\mathbf{x}}_i^{t,\tau-1} - \bar{\mathbf{x}}^t - \eta_c \frac{1}{n_i} \sum_{j\in\mathcal{S}_i^{t,\tau-1}} \sum_{h=0}^{I_i} \eta \mathbf{g}_j^{t,\tau-1,h} \right\|^2$$

$$= \mathbb{E}_t \left\| \bar{\mathbf{x}}_i^{t,\tau-1} - \bar{\mathbf{x}}^t - \eta_c \frac{1}{n_i} \sum_{j\in\mathcal{S}_i^{t,\tau-1}} \sum_{h=0}^{I_i} \eta \left( \mathbf{g}_j^{t,\tau-1,h} - \nabla F_j(\mathbf{x}_j^{t,\tau-1,h}) + \nabla F_j(\mathbf{x}_j^{t,\tau-1,h}) \right) \right.$$

$$+ \frac{1}{m_i} \sum_{j\in\mathcal{V}_i} \sum_{h=0}^{I_i} \eta \left( \nabla F_j(\mathbf{x}_j^{t,\tau-1,h}) - \nabla F_j(\mathbf{x}_j^{t,\tau-1,h}) - \nabla f_i(\bar{\mathbf{x}}_i^{t,\tau-1}) + \nabla f_i(\bar{\mathbf{x}}_i^{t,\tau-1}) \right.$$

$$\left. \left. - \nabla f_i(\bar{\mathbf{x}}^t) + \nabla f_i(\bar{\mathbf{x}}^t) - \nabla f(\bar{\mathbf{x}}^t) + \nabla f(\bar{\mathbf{x}}^t) \right) \right\|^2$$

$$\overset{(a)}{=} \eta_c^2\eta^2 \mathbb{E}_t \left\| \frac{1}{n_i} \sum_{j\in\mathcal{S}_i^{t,\tau-1}} \sum_{h=0}^{I_i} \eta \left( \mathbf{g}_j^{t,\tau-1,h} - \nabla F_j(\mathbf{x}_j^{t,\tau-1,h}) \right) \right\|^2$$

$$+ \eta_c^2\eta^2 \mathbb{E}_t \left\| \frac{1}{n_i} \sum_{j\in\mathcal{S}_i^{t,\tau-1}} \sum_{h=0}^{I_i} \nabla F_j(\mathbf{x}_j^{t,\tau-1,h}) - \frac{1}{m_i} \sum_{j\in\mathcal{V}_i} \sum_{h=0}^{I_i} \nabla F_j(\mathbf{x}_j^{t,\tau-1,h}) \right\|^2$$

$$+ \mathbb{E}_t \left\| \bar{\mathbf{x}}_i^{t,\tau-1} - \bar{\mathbf{x}}^t - \eta_c \frac{1}{m_i} \sum_{j\in\mathcal{V}_i} \sum_{h=0}^{I_i} \eta \left( \nabla F_j(\mathbf{x}_j^{t,\tau-1,h}) - \nabla F_j(\bar{\mathbf{x}}_i^{t,\tau-1}) \right) \right.$$

$$\left. + \nabla f_i(\bar{\mathbf{x}}_i^{t,\tau-1}) - \nabla f_i(\bar{\mathbf{x}}^t) + \nabla f_i(\bar{\mathbf{x}}^t) - \nabla f(\bar{\mathbf{x}}^t) + \nabla f(\bar{\mathbf{x}}^t) \right) \right\|^2$$

$$\overset{(b)}{\le} \frac{I_i\eta_c^2\eta^2\sigma^2}{n_i} + \eta_c^2\eta^2 \mathbb{E}_t \left\| \frac{1}{n_i} \sum_{j\in\mathcal{S}_i^{t,\tau-1}} \sum_{h=0}^{I_i} \nabla F_j(\mathbf{x}_j^{t,\tau-1,h}) - \frac{1}{m_i} \sum_{j\in\mathcal{V}_i} \sum_{h=0}^{I_i} \nabla F_j(\mathbf{x}_j^{t,\tau-1,h}) \right\|^2$$

$$+ (1 + \frac{1}{2\omega_i-1})\mathbb{E}_t \left\| \bar{\mathbf{x}}_i^{t,\tau-1} - \bar{\mathbf{x}}^t \right\|^2 + 8\omega_i\eta_c^2\eta^2 \left( \mathbb{E}_t \left\| \frac{1}{m_i} \sum_{j\in\mathcal{V}_i} \sum_{h=0}^{I_i} \left( \nabla F_j(\mathbf{x}_j^{t,\tau-1,h}) - \nabla F_j(\bar{\mathbf{x}}_i^{t,\tau-1}) \right) \right\|^2 \right.$$

$$\left. + I_i^2 \mathbb{E}_t \left\| \nabla f_i(\bar{\mathbf{x}}_i^{t,\tau-1}) - \nabla f_i(\bar{\mathbf{x}}^t) \right\|^2 + I_i^2 \mathbb{E}_t \left\| \nabla f_i(\bar{\mathbf{x}}^t) - \nabla f(\bar{\mathbf{x}}^t) \right\|^2 + I_i^2 \mathbb{E}_t \left\| \nabla f(\bar{\mathbf{x}}^t) \right\|^2 \right)$$

$$\leq \frac{I_i \eta_c^2 \eta^2 \sigma^2}{n_i} + \eta_c^2 \eta^2 \mathbb{E}_t \left\| \frac{1}{n_i} \sum_{j \in \mathcal{S}_i^{t,\tau-1}} \sum_{h=0}^{I_i} \nabla F_j(\mathbf{x}_j^{t,\tau-1,h}) - \frac{1}{m_i} \sum_{j \in \mathcal{V}_i} \sum_{h=0}^{I_i} \nabla F_j(\mathbf{x}_j^{t,\tau-1,h}) \right\|^2$$

$$+ 8G\eta_c^2 \eta^2 L^2 \frac{1}{m_i} \sum_{j \in \mathcal{V}_i} \sum_{h=0}^{I_i} \mathbb{E}_t \left\| \mathbf{x}_j^{t,\tau-1,h} - \bar{\mathbf{x}}_i^{t,\tau-1} \right\|^2 + 8G I_i \eta_c^2 \eta^2 \epsilon^2$$

$$+ (1 + \frac{1}{2\omega_i - 1} + 8G I_i \eta_c^2 \eta^2 L^2) \mathbb{E}_t \left\| \bar{\mathbf{x}}_i^{t,\tau-1} - \bar{\mathbf{x}}^t \right\|^2 + 8G I_i \eta_c^2 \eta^2 \left\| \nabla f(\bar{\mathbf{x}}^t) \right\|^2, \tag{12}$$

where $(a)$ holds due to the zero mean and independence of the first two term, and the definition $f_i(\bar{\mathbf{x}}_i^{t,\tau}) = \frac{1}{m_i} \sum_{j \in \mathcal{V}_i} F_j(\bar{\mathbf{x}}_i^{t,\tau})$ is also used for substitution in the last term. The first term of $(b)$ is acquired via Lemma 2 and bounded variance (Assumption 3). The other terms of $(b)$ holds due to the fact that $||x + y||^2 \leq (1 + \frac{1}{k-1})||x||^2 + k||y||^2, \forall k > 1$ (we set $k = 2\omega_i$ here).

For better presentation, we denote $\Psi_i^{t,\tau} = \mathbb{E} \| \frac{1}{n_i} \sum_{j \in \mathcal{S}_i^{t,\tau}} \sum_{h=0}^{I_i} \nabla F_j(\mathbf{x}_j^{t,\tau,h}) - \frac{1}{m_i} \sum_{j \in \mathcal{V}_i} \sum_{h=0}^{I_i} \nabla F_j(\mathbf{x}_j^{t,\tau,h}) \|^2$ and $\Omega_i^{t,\tau} = \frac{1}{m_i} \sum_{j \in \mathcal{V}_i} \sum_{h=0}^{I_i} \mathbb{E} \| \mathbf{x}_j^{t,\tau,h} - \bar{\mathbf{x}}_i^{t,\tau} \|^2$. Note that $\Psi_i^{t,\tau}$ is essentially a measurement of the additional variance introduced by worker sampling in cluster round $\tau$ of cluster $i$. Suppose $\eta_c \eta \leq \frac{1}{8LG}$, we can have

$$\mathbb{E} \left\| \bar{\mathbf{x}}_i^{t,\tau} - \bar{\mathbf{x}}^t \right\|^2 \leq (1 + \frac{1}{2\omega_i - 1} + 8G I_i \eta_c^2 \eta^2 L^2) \mathbb{E} \left\| \bar{\mathbf{x}}_i^{t,\tau-1} - \bar{\mathbf{x}}^t \right\|^2 + \frac{I_i \eta_c^2 \eta^2 \sigma^2}{n_i} + \eta_c^2 \eta^2 \Psi_i^{t,\tau-1}$$

$$+ 8G I_i \eta_c^2 \eta^2 \epsilon^2 + 8G \eta_c^2 \eta^2 L^2 \Omega_i^{t,\tau-1} + 8G I_i \eta_c^2 \eta^2 \left\| \nabla f(\bar{\mathbf{x}}^t) \right\|^2$$

$$\leq (1 + \frac{1}{\omega_i - 1}) \mathbb{E} \left\| \bar{\mathbf{x}}_i^{t,\tau-1} - \bar{\mathbf{x}}^t \right\|^2 + \frac{I_i \eta_c^2 \eta^2 \sigma^2}{n_i} + 8G I_i \eta_c^2 \eta^2 \epsilon^2 + 8G I_i \eta_c^2 \eta^2 \left\| \nabla f(\bar{\mathbf{x}}^t) \right\|^2$$

$$+ \eta_c^2 \eta^2 \Psi_i^{t,\tau-1} + 8G \eta_c^2 \eta^2 L^2 \Omega_i^{t,\tau-1}. \tag{13}$$

Unrolling the recursion, we obtain

$$\mathbb{E} \left\| \bar{\mathbf{x}}_i^{t,\tau} - \bar{\mathbf{x}}^t \right\|^2$$

$$\leq \sum_{p=0}^{\tau-1} (1 + \frac{1}{\omega_i - 1})^p \left[ \frac{I_i \eta_c^2 \eta^2 \sigma^2}{n_i} + 8G I_i \eta_c^2 \eta^2 \epsilon^2 + 8G I_i \eta_c^2 \eta^2 \left\| \nabla f(\bar{\mathbf{x}}^t) \right\|^2 \right.$$

$$\left. + \eta_c^2 \eta^2 \Psi_i^{t,\tau-1-p} + 8G \eta_c^2 \eta^2 L^2 \Omega_i^{t,\tau-1-p} \right]$$

$$\leq (\omega_i - 1) \left[ (1 + \frac{1}{\omega_i - 1})^{\omega_i} - 1 \right] \left[ \frac{I_i \eta_c^2 \eta^2 \sigma^2}{n_i} + 8G I_i \eta_c^2 \eta^2 \epsilon^2 + 8G I_i \eta_c^2 \eta^2 \left\| \nabla f(\bar{\mathbf{x}}^t) \right\|^2 \right]$$

$$+ (1 + \frac{1}{\omega_i - 1})^{\omega_i - 1} \left[ \eta_c^2 \eta^2 \sum_{p=0}^{\tau-1} \Psi_i^{t,p} + 8G \eta_c^2 \eta^2 L^2 \sum_{p=0}^{\tau-1} \Omega_i^{t,p} \right]$$

$$\overset{(a)}{\leq} 5\omega_i \left[ \frac{I_i \eta_c^2 \eta^2 \sigma^2}{n_i} + 8G I_i \eta_c^2 \eta^2 \epsilon^2 + 8G I_i \eta_c^2 \eta^2 \left\| \nabla f(\bar{\mathbf{x}}^t) \right\|^2 \right] + 3\eta_c^2 \eta^2 \sum_{p=0}^{\tau-1} \Psi_i^{t,p} + 24G \eta_c^2 \eta^2 L^2 \sum_{p=0}^{\tau-1} \Omega_i^{t,p}$$

$$= \frac{5G \eta_c^2 \eta^2 \sigma^2}{n_i} + 40G^2 \eta_c^2 \eta^2 \epsilon^2 + 40G^2 \eta_c^2 \eta^2 \left\| \nabla f(\bar{\mathbf{x}}^t) \right\|^2 + 3\eta_c^2 \eta^2 \sum_{p=0}^{\tau-1} \Psi_i^{t,p} + 24G \eta_c^2 \eta^2 L^2 \sum_{p=0}^{\tau-1} \Omega_i^{t,p}, \tag{14}$$

where $(a)$ is due to the fact that $(1 + \frac{1}{\omega_i-1})^{\omega_i - 1} \leq 3$ and $(1 + \frac{1}{\omega_i-1})^{\omega_i} \leq 5$ for $\omega_i > 1$.

Summing from $\tau = 0, \ldots, \omega_i - 1$, the overall cluster-master parameter MSE of cluster $i$ in a master round $t$ can be expressed as

$$\sum_{\tau=0}^{\omega_i-1} \mathbb{E}_t \left\| \bar{\mathbf{x}}_i^{t,\tau} - \bar{\mathbf{x}}^t \right\|^2 \overset{(a)}{\leq} 5G \eta_c^2 \eta^2 \frac{\omega_i}{n_i} \sigma^2 + 40\omega_i G^2 \eta_c^2 \eta^2 \epsilon^2$$

$$+ 40\omega_i G^2 \eta_c^2 \eta^2 \left\| \nabla f(\bar{\mathbf{x}}^t) \right\|^2 + 24G \eta_c^2 \eta^2 L^2 \omega_i \sum_{\tau=0}^{\omega_i-1} \Omega_i^{t,\tau} + 3\eta_c^2 \eta^2 \omega_i \sum_{\tau=0}^{\omega_i-1} \Psi_i^{t,\tau}. \tag{15}$$

where the above simplification follows $\sum_{\tau=0}^{\omega_i-1}\sum_{p=0}^{\tau-1}\Omega_i^{t,p} \le \sum_{\tau=0}^{\omega_i-1}\sum_{p=0}^{\tau}\Omega_i^{t,p} \le \omega_i \sum_{\tau=0}^{\omega_i-1}\Omega_i^{t,\tau}$, and the same holds for $\Psi_i^{t,\tau}$.

Note that when degenerating to full participation case, we could set $n_i = m_i$ and $\Psi_i^{t,\tau} = 0, \forall\tau$, and thereby recover the corresponding result in Lemma 3. This concludes the proof. $\qquad\square$

## C.2 BOUNDING WORKER-CLUSTER PARAMETER MSE

Then we bound the worker-cluster parameter MSE. Here we consider the intra-cluster aggregation process as a standard FL process, and introduce two lemmas here to bound it.

**Lemma 4** (Generalization of Lemma 3 from Reddi et al. [2020]). *For any local learning rate $\eta \le \frac{1}{8I_iL}$, we have the following result for cluster $i$ in its cluster round $\tau$*

$$\frac{1}{m_i}\sum_{j\in\mathcal{V}_i}\mathbb{E}_t\big\|\mathbf{x}_j^{t,\tau,h} - \bar{\mathbf{x}}_i^{t,\tau}\big\|^2 \le 5I_i\eta^2(\sigma^2 + 10I_i\epsilon_i^2) + 50I_i^2\eta^2\epsilon^2$$
$$+ 50I_i^2\eta^2L^2\mathbb{E}_t\|\bar{\mathbf{x}}_i^{t,\tau} - \bar{\mathbf{x}}^t\|^2 + 50I_i^2\eta^2\|\nabla f(\bar{\mathbf{x}}^t)\|^2. \tag{16}$$

*Proof.* Our proof here is a variant of that of Reddi et al. [2020] in the HFL. For any worker $j$ in cluster $i$, we have for any local step $h$,

$$\mathbb{E}_t\big\|\mathbf{x}_j^{t,\tau,h} - \bar{\mathbf{x}}_i^{t,\tau}\big\|^2 = \mathbb{E}\big\|\mathbf{x}_j^{t,\tau,h-1} - \bar{\mathbf{x}}_i^{t,\tau} - \eta\mathbf{g}_j^{t,\tau,h-1}\big\|^2$$

$$= \mathbb{E}_t\Big\|\mathbf{x}_j^{t,\tau,h-1} - \bar{\mathbf{x}}_i^{t,\tau} - \eta\Big(\mathbf{g}_j^{t,\tau,h-1} - \nabla F_j(\mathbf{x}_j^{t,\tau,h-1}) + \nabla F_j(\mathbf{x}_j^{t,\tau,h-1}) - \nabla F_i(\bar{\mathbf{x}}_i^{t,\tau})$$

$$+ \nabla F_i(\bar{\mathbf{x}}_i^{t,\tau}) - \nabla f_i(\bar{\mathbf{x}}_i^{t,\tau}) + \nabla f_i(\bar{\mathbf{x}}_i^{t,\tau}) - \nabla f_i(\bar{\mathbf{x}}^t) + \nabla f_i(\bar{\mathbf{x}}^t) - \nabla f(\bar{\mathbf{x}}^t) + \nabla f(\bar{\mathbf{x}}^t)\Big)\Big\|^2$$

$$\overset{(a)}{\le} (1 + \frac{1}{2I_i-1})\mathbb{E}_t\big\|\mathbf{x}_j^{t,\tau,h-1} - \bar{\mathbf{x}}_i^{t,\tau}\big\|^2 + \eta^2\mathbb{E}_t\big\|\mathbf{g}_j^{t,\tau,h-1} - \nabla F_j(\mathbf{x}_j^{t,\tau,h-1})\big\|^2$$

$$+ 10I_i\eta^2\mathbb{E}_t\big\|\nabla F_j(\mathbf{x}_j^{t,\tau,h-1}) - \nabla F_j(\bar{\mathbf{x}}_i^{t,\tau})\big\|^2 + 10I_i\eta^2\mathbb{E}_t\big\|\nabla F_j(\bar{\mathbf{x}}_i^{t,\tau}) - \nabla f_i(\bar{\mathbf{x}}_i^{t,\tau})\big\|^2$$

$$+ 10I_i\eta^2\mathbb{E}_t\big\|\nabla f_i(\bar{\mathbf{x}}_i^{t,\tau}) - \nabla f_i(\bar{\mathbf{x}}^t)\big\|^2 + 10I_i\eta^2\mathbb{E}_t\big\|\nabla f_i(\bar{\mathbf{x}}^t) - \nabla f(\bar{\mathbf{x}}^t)\big\|^2 + 10I_i\eta^2\mathbb{E}_t\big\|\nabla f(\bar{\mathbf{x}}^t)\big\|^2$$

$$\le (1 + \frac{1}{2I_i-1} + 10I_i\eta^2L^2)\mathbb{E}_t\big\|\mathbf{x}_j^{t,\tau,h-1} - \bar{\mathbf{x}}_i^{t,\tau}\big\|^2 + \eta^2\sigma^2 + 10I_i\eta^2\epsilon_i^2$$

$$+ 10I_i\eta^2L^2\mathbb{E}_t\big\|\bar{\mathbf{x}}_i^{t,\tau} - \bar{\mathbf{x}}^t\big\|^2 + 10I_i\eta^2\epsilon^2 + 10I_i\eta^2\mathbb{E}_t\big\|\nabla f(\bar{\mathbf{x}}^t)\big\|^2$$

$$\le (1 + \frac{1}{I_i-1})\mathbb{E}_t\big\|\mathbf{x}_j^{t,\tau,h-1} - \bar{\mathbf{x}}_i^{t,\tau}\big\|^2 + \eta^2\sigma^2 + 10I_i\eta^2\epsilon_i^2 + 10I_i\eta^2\epsilon^2$$

$$+ 10I_i\eta^2L^2\mathbb{E}_t\big\|\bar{\mathbf{x}}_i^{t,\tau} - \bar{\mathbf{x}}^t\big\|^2 + 10I_i\eta^2\mathbb{E}_t\big\|\nabla f(\bar{\mathbf{x}}^t)\big\|^2 \tag{17}$$

where the expansion of $(a)$ holds similarly as we prove Lemma 3, i.e., $\|x + y\|^2 \le (1 + \frac{1}{k-1})\|x\|^2 + k\|y\|^2, \forall k > 1$ with $k = 2I_i$.

Unrolling the recursion, we get

$$\frac{1}{m_i}\sum_{j\in\mathcal{V}_i}\mathbb{E}_t\big\|\mathbf{x}_j^{t,\tau,h} - \bar{\mathbf{x}}_i^{t,\tau}\big\|^2$$

$$\le (I_i - 1)\Big[(1 + \frac{1}{I_i-1})^{I_i} - 1\Big]\Big[\eta^2\sigma^2 + 10I_i\eta^2\epsilon_i^2 + 10I_i\eta^2\epsilon^2$$

$$+ 10I_i\eta^2L^2\mathbb{E}_t\big\|\bar{\mathbf{x}}_i^{t,\tau} - \bar{\mathbf{x}}^t\big\|^2 + 10I_i\eta^2\mathbb{E}_t\big\|\nabla f(\bar{\mathbf{x}}^t)\big\|^2\Big]$$

$$\overset{(a)}{\le} 5I_i\eta^2(\sigma^2 + 10I_i\epsilon_i^2) + 50I_i^2\eta^2L^2\mathbb{E}_t\big\|\bar{\mathbf{x}}_i^{t,\tau} - \bar{\mathbf{x}}^t\big\|^2 + 50I_i^2\eta^2\epsilon^2 + 50I_i^2\eta^2\|\nabla f(\bar{\mathbf{x}}^t)\|^2, \tag{18}$$

where $(a)$ is due to the fact that $(1 + \frac{1}{I_i-1})^{I_i} \le 5$ for $I_i > 1$. This completes the proof. $\qquad\square$

Note that Lemma 3 and Lemma 4 indicate that the cluster-master parameter MSE and worker-cluster parameter MSE can be bounded via each other. We thus utilize these two lemmas with proper learning rate condition to derive a more general lemma for worker-cluster parameter MSE, which is not depends on the cluster-master one.

**Lemma 5.** *For any local learning rate $\eta$ and cluster learning rate $\eta_c$ satisfying $\eta_c\eta \leq \frac{1}{10LG}$ and $\eta \leq \frac{1}{10I_{max}L}$, where $I_{max} = \max_i I_i$, we can bound the overall worker-cluster parameter MSE in a master round $t$ with arbitrary positive coefficient $p_i > 0, \forall i \in [M]$ as, with full worker participation,*

$$\sum_{i=1}^{M} p_i \sum_{\tau=0}^{\omega_i-1} \sum_{j\in\mathcal{V}_i} \sum_{h=0}^{I_i-1} \mathbb{E}_t\|\mathbf{x}_j^{t,\tau,h} - \bar{\mathbf{x}}_i^{t,\tau}\|^2 \leq 6G\eta^2 \sum_{i=1}^{M} p_i m_i I_i \sigma^2 + 60G\eta^2 \sum_{i=1}^{M} p_i m_i I_i^2 \epsilon_i^2$$

$$+ 80G\eta^2 \sum_{i=1}^{M} p_i m_i I_i^2 \|\nabla f(\bar{\mathbf{x}}^t)\|^2 + 80G\eta^2 \sum_{i=1}^{M} p_i m_i I_i^2 \epsilon^2 + 3G^2\eta_c^2\eta^2 \sum_{i=1}^{M} p_i \sigma^2. \tag{19}$$

*With partial worker participation for both sampling strategies, we have*

$$\sum_{i=1}^{M} p_i \sum_{\tau=0}^{\omega_i-1} \sum_{j\in\mathcal{V}_i} \sum_{h=0}^{I_i-1} \mathbb{E}_t\|\mathbf{x}_j^{t,\tau,h} - \bar{\mathbf{x}}_i^{t,\tau}\|^2 \leq 6G\eta^2 \sum_{i=1}^{M} p_i m_i I_i \sigma^2 + 60G\eta^2 \sum_{i=1}^{M} p_i m_i I_i^2 \epsilon_i^2$$

$$+ 80G\eta^2 \sum_{i=1}^{M} p_i m_i I_i^2 \|\nabla f(\bar{\mathbf{x}}^t)\|^2 + 80G\eta^2 \sum_{i=1}^{M} p_i m_i I_i^2 \epsilon^2$$

$$+ 3G^2\eta_c^2\eta^2 \sum_{i=1}^{M} \frac{p_i m_i}{n_i}\sigma^2 + 2G\eta_c^2\eta^2 \sum_{i=1}^{M} p_i m_i \sum_{\tau=0}^{\omega_i-1} \Psi_i^{t,\tau}. \tag{20}$$

*where $\Psi_i^{t,\tau} = \mathbb{E}_t\|\frac{1}{n_i}\sum_{j\in\mathcal{S}_i^{t,\tau}}\sum_{h=0}^{I_i}\nabla F_j(\mathbf{x}_j^{t,\tau,h}) - \frac{1}{m_i}\sum_{j\in\mathcal{V}_i}\sum_{h=0}^{I_i}\nabla F_j(\mathbf{x}_j^{t,\tau,h})\|^2.$*

*Proof.* With Lemma 4, we have for arbitrary $p_i > 0, \forall i \in [M]$,

$$\sum_{i=1}^{M} p_i \sum_{\tau=0}^{\omega_i-1} \sum_{h=0}^{I_i-1} \sum_{j\in\mathcal{V}_i} \mathbb{E}_t\|\mathbf{x}_j^{t,\tau,h} - \bar{\mathbf{x}}_i^{t,\tau}\|^2$$

$$\leq 5G\eta^2 \sum_{i=1}^{M} p_i m_i I_i \sigma^2 + 50G\eta^2 \sum_{i=1}^{M} p_i m_i I_i^2 \epsilon_i^2 + 50G\eta^2 \sum_{i=1}^{M} p_i m_i I_i^2 \|\nabla f(\bar{\mathbf{x}}^t)\|^2$$

$$+ 50G\eta^2 \sum_{i=1}^{M} p_i m_i I_i^2 \epsilon^2 + 50\eta^2 L^2 \sum_{i=1}^{M} p_i m_i I_i^3 \sum_{\tau=0}^{\omega_i-1} \mathbb{E}_t\|\bar{\mathbf{x}}_i^{t,\tau} - \bar{\mathbf{x}}^t\|^2, \tag{21}$$

With Lemma 3, we have for the last term of Eq. 21 as

$$50\eta^2 L^2 \sum_{i=1}^{M} p_i m_i I_i^3 \sum_{\tau=0}^{\omega_i-1} \mathbb{E}_t\|\bar{\mathbf{x}}_i^{t,\tau} - \bar{\mathbf{x}}^t\|^2$$

$$\overset{(a)}{\leq} \frac{5}{2}G^2\eta_c^2\eta^2 \sum_{i=1}^{M} \frac{p_i m_i}{n_i}\sigma^2 + 20G\eta^2 \sum_{i=1}^{M} p_i m_i I_i^2 \epsilon^2 + 20G\eta^2 \sum_{i=1}^{M} p_i m_i I_i^2 \|\nabla f(\bar{\mathbf{x}}^t)\|^2$$

$$+ \frac{1}{8}\sum_{i=1}^{M} p_i \sum_{\tau=0}^{\omega_i-1} \sum_{j\in\mathcal{V}_i} \sum_{h=0}^{I_i-1} \mathbb{E}_t\|\mathbf{x}_j^{t,\tau,h} - \bar{\mathbf{x}}_i^{t,\tau}\|^2 + \frac{3}{2}G\eta_c^2\eta^2 \sum_{i=1}^{M} p_i m_i \sum_{\tau=0}^{\omega_i-1} \Psi_i^{t,\tau}, \tag{22}$$

where in $(a)$, we slightly tighten the condition as $\eta_c\eta \leq \frac{1}{10LG}$ for the middle three terms. We use disparate treatment $\eta \leq \frac{1}{10I_{max}L}$ for the first term since it can be further merged to some term in $A_2$. For the last term, both two conditions are applicable as we will show in bounding $A_2$, so here we choose a simpler one, i.e., $\eta \leq \frac{1}{10I_{max}L}$. Note that both these two conditions for simplification will not incur any fundamental changes in the convergence behavior, since we can further merge them into some lower-order terms. This could also provide much more readability.

Substituting Eq. 22 back into Eq. 21, merging the left hand term, we have

$$
\sum_{i=1}^{M} p_i \sum_{\tau=0}^{\omega_i-1} \sum_{j \in \mathcal{V}_i} \sum_{h=0}^{I_i-1} \mathbb{E}_t \| \mathbf{x}_j^{t,\tau,h} - \bar{\mathbf{x}}_i^{t,\tau} \|^2
$$

$$
\leq 6G\eta^2 \sum_{i=1}^{M} p_i m_i I_i \sigma^2 + 60G\eta^2 \sum_{i=1}^{M} p_i m_i I_i^2 \epsilon_i^2 + 80G\eta^2 \sum_{i=1}^{M} p_i m_i I_i^2 \| \nabla f(\bar{\mathbf{x}}^t) \|^2
$$

$$
+ 80G\eta^2 \sum_{i=1}^{M} p_i m_i I_i^2 \epsilon^2 + 3G^2 \eta_c^2 \eta^2 \sum_{i=1}^{M} \frac{p_i m_i}{n_i} \sigma^2 + 2G\eta_c^2 \eta^2 \sum_{i=1}^{M} p_i m_i \sum_{\tau=0}^{\omega_i-1} \Psi_i^{t,\tau}. \tag{23}
$$

Similarly, for full participation case, we set $n_i = m_i$, $\Psi_i^{t,\tau} = 0$, $\forall i \in [M], \tau$ and thus recover the corresponding result in Lemma 5. $\square$

With Eq. 9, Lemma 3, and Lemma 5 for full participation, and setting $p_i = \frac{1}{m}$ in Lemma 5, we rearrange Eq. 8 to bound for $-A_1$ as

$$
-A_1 \leq \frac{\eta_c \eta G}{2} \| \nabla f(\bar{\mathbf{x}}^t) \|^2 - \frac{\eta_c \eta}{2G} \mathbb{E}_t \Big\| \sum_{i=1}^{M} \frac{m_i}{m} \sum_{\tau=0}^{\omega_i-1} \sum_{h=0}^{I_i-1} \frac{1}{m_i} \sum_{j \in \mathcal{V}_i} \nabla F_j(\mathbf{x}_j^{t,\tau,h}) \Big\|^2
$$

$$
+ L^2 \eta_c \eta \Big( 5G^2 \eta_c^2 \eta^2 \frac{M}{m} \sigma^2 + 40G^3 \eta_c^2 \eta^2 \epsilon^2 + 40G^3 \eta_c^2 \eta^2 \| \nabla f(\bar{\mathbf{x}}^t) \|^2 + \frac{124}{100} \Big( 6G\eta^2 \sum_{i=1}^{M} \frac{m_i}{m} I_i \sigma^2
$$

$$
+ 60G\eta^2 \sum_{i=1}^{M} \frac{m_i}{m} I_i^2 \epsilon_i^2 + 80G\eta^2 \sum_{i=1}^{M} \frac{m_i}{m} I_i^2 \| \nabla f(\bar{\mathbf{x}}^t) \|^2 + 80G\eta^2 \sum_{i=1}^{M} \frac{m_i}{m} I_i^2 \epsilon^2 + 3G^2 \eta_c^2 \eta^2 \frac{M}{m} \sigma^2 \Big) \Big)
$$

$$
\leq \frac{\eta_c \eta G}{2} \| \nabla f(\bar{\mathbf{x}}^t) \|^2 - \frac{\eta_c \eta}{2G} \mathbb{E}_t \Big\| \sum_{i=1}^{M} \frac{m_i}{m} \sum_{\tau=0}^{\omega_i-1} \sum_{h=0}^{I_i-1} \frac{1}{m_i} \sum_{j \in \mathcal{V}_i} \nabla F_j(\mathbf{x}_j^{t,\tau,h}) \Big\|^2
$$

$$
+ 9G^2 \eta_c^3 \eta^3 L^2 \frac{M}{m} \sigma^2 + 40G^3 \eta_c^3 \eta^3 L^2 \epsilon^2 + 40G^3 \eta_c^3 \eta^3 L^2 \| \nabla f(\bar{\mathbf{x}}^t) \|^2 + 8G\eta_c \eta^3 L^2 \sum_{i=1}^{M} \frac{m_i}{m} I_i \sigma^2
$$

$$
+ 75G\eta_c \eta^3 L^2 \sum_{i=1}^{M} \frac{m_i}{m} I_i^2 \epsilon_i^2 + 100G\eta_c \eta^3 L^2 \sum_{i=1}^{M} \frac{m_i}{m} I_i^2 \| \nabla f(\bar{\mathbf{x}}^t) \|^2 + 100G\eta_c \eta^3 L^2 \sum_{i=1}^{M} \frac{m_i}{m} I_i^2 \epsilon^2, \tag{24}
$$

For the term $A_2$, we have

$$
A_2 = \mathbb{E}_t \Big\| \sum_{i=1}^{M} \frac{m_i}{m} \sum_{\tau=0}^{\omega_i-1} \eta_c \sum_{h=0}^{I_i-1} \frac{1}{m_i} \sum_{j \in \mathcal{V}_i} \eta \mathbf{g}_j^{t,\tau,h} \Big\|^2
$$

$$
\overset{(a)}{=} \eta_c^2 \eta^2 \mathbb{E}_t \Big\| \sum_{i=1}^{M} \frac{m_i}{m} \sum_{\tau=0}^{\omega_i-1} \sum_{h=0}^{I_i-1} \frac{1}{m_i} \sum_{j \in \mathcal{V}_i} \big( \mathbf{g}_j^{t,\tau,h} - \nabla F_j(\mathbf{x}_j^{t,\tau,h}) \big) \Big\|^2
$$

$$
+ \eta_c^2 \eta^2 \mathbb{E}_t \Big\| \sum_{i=1}^{M} \frac{m_i}{m} \sum_{\tau=0}^{\omega_i-1} \sum_{h=0}^{I_i-1} \frac{1}{m_i} \sum_{j \in \mathcal{V}_i} \nabla F_j(\mathbf{x}_j^{t,\tau,h}) \Big\|^2
$$

$$
\leq \frac{G\eta_c^2 \eta^2}{m} \sigma^2 + \eta_c^2 \eta^2 \mathbb{E}_t \Big\| \sum_{i=1}^{M} \frac{m_i}{m} \sum_{\tau=0}^{\omega_i-1} \sum_{h=0}^{I_i-1} \frac{1}{m_i} \sum_{j \in \mathcal{V}_i} \nabla F_j(\mathbf{x}_j^{t,\tau,h}) \Big\|^2, \tag{25}
$$

where $(a)$ follows the fact that $\mathbb{E} \| \mathbf{x} \|^2 = \mathbb{E} \| \mathbf{x} - \mathbb{E}\mathbf{x} \|^2 + \| \mathbb{E}\mathbf{x} \|^2$.

Substituting Eq. 24 and Eq. 25 back into Eq. 7 and rearranging the order, we have

$$
\eta_g \eta_c \eta G \Big( \frac{1}{2} - 40G^2 \eta_c^2 \eta^2 L^2 - 100\eta^2 L^2 \sum_{i=1}^{M} \frac{m_i}{m} I_i^2 \Big) \| \nabla f(\bar{\mathbf{x}}^t) \|^2
$$

$$\leq f(\overline{\mathbf{x}}^t) - \mathbb{E}_t[f(\overline{\mathbf{x}}^{t+1})] + \left(\frac{L\eta_g^2\eta_c^2\eta^2}{2} - \frac{\eta_g\eta_c\eta}{2G}\right)\mathbb{E}_t\left\|\frac{1}{m}\sum_{i=1}^{M}\sum_{\tau=0}^{\omega_i-1}\sum_{h=0}^{I_i-1}\sum_{j\in\mathcal{V}_i}\nabla F_j(\mathbf{x}_j^{t,\tau,h})\right\|^2$$

$$+\frac{GL\eta_g^2\eta_c^2\eta^2}{2m}\sigma^2 + 9G^2\eta_g\eta_c^3\eta^3L^2\frac{M}{m}\sigma^2 + 40G^3\eta_g\eta_c^3\eta^3L^2\epsilon^2 + 8G\eta_g\eta_c\eta^3L^2\sum_{i=1}^{M}\frac{m_i}{m}I_i\sigma^2$$

$$+75G\eta_g\eta_c\eta^3L^2\sum_{i=1}^{M}\frac{m_i}{m}I_i^2\epsilon_i^2 + 100G\eta_g\eta_c\eta^3L^2\sum_{i=1}^{M}\frac{m_i}{m}I_i^2\epsilon^2$$

$$\overset{(a)}{\leq} f(\overline{\mathbf{x}}^t) - \mathbb{E}_t[f(\overline{\mathbf{x}}^{t+1})] + \frac{GL\eta_g^2\eta_c^2\eta^2}{2m}\sigma^2 + 9G^2\eta_g\eta_c^3\eta^3L^2\frac{M}{m}\sigma^2 + 8G\eta_g\eta_c\eta^3L^2\sum_{i=1}^{M}\frac{m_i}{m}I_i\sigma^2$$

$$+75G\eta_g\eta_c\eta^3L^2\sum_{i=1}^{M}\frac{m_i}{m}I_i^2\epsilon_i^2 + 100G\eta_g\eta_c\eta^3L^2\sum_{i=1}^{M}\frac{m_i}{m}I_i^2\epsilon^2 + 40G^3\eta_g\eta_c^3\eta^3L^2\epsilon^2, \tag{26}$$

where $(a)$ follows from $\frac{L\eta_g^2\eta_c^2\eta^2}{2} - \frac{\eta_g\eta_c\eta}{2G} \leq 0$ if $\eta_g\eta_c\eta \leq \frac{1}{GL}$.

Suppose $40G^2\eta_c^2\eta^2L^2 + 100\eta^2L^2\sum_{i=1}^{M}\frac{m_i}{m}I_i^2 < \frac{1}{2}$, and there exists a constant satisfying $(\frac{1}{2} - 40G^2\eta_c^2\eta^2L^2 - 100\eta^2L^2\sum_{i=1}^{M}\frac{m_i}{m}I_i^2) > c > 0$, then we have

$$\left\|\nabla f(\overline{\mathbf{x}}^t)\right\|^2 \leq \frac{f(\overline{\mathbf{x}}^t) - \mathbb{E}_t[f(\overline{\mathbf{x}}^{t+1})]}{\eta_g\eta_c\eta Gc} + \frac{1}{c}\left[\frac{L\eta_g\eta_c\eta}{2m}\sigma^2 + 9G\eta_c^2\eta^2L^2\frac{M}{m}\sigma^2 + 8\eta^2L^2\sum_{i=1}^{M}\frac{m_i}{m}I_i\sigma^2\right.$$

$$\left. +75\eta^2L^2\sum_{i=1}^{M}\frac{m_i}{m}I_i^2\epsilon_i^2 + 100\eta^2L^2\sum_{i=1}^{M}\frac{m_i}{m}I_i^2\epsilon^2 + 40G^2\eta_c^2\eta^2L^2\epsilon^2\right], \tag{27}$$

Taking a double expectation over the data samples among all workers and averaging from $t = 0, 1, \ldots, T$, we have the final results as

$$\min_{t\in[T]}\mathbb{E}\left\|\nabla f(\overline{\mathbf{x}}^t)\right\|^2 \leq \frac{f_0 - f_*}{c\eta_g\eta_c\eta GT} + \frac{1}{c}(\Phi_1 + \Phi_2), \tag{28}$$

where

$$\Phi_1 = 9G\eta_c^2\eta^2L^2\frac{M}{m}\sigma^2 + 8\eta^2L^2\sum_{i=1}^{M}\frac{m_i}{m}I_i\sigma^2 + 40G^2\eta_c^2\eta^2L^2\epsilon^2,$$

$$+75\eta^2L^2\sum_{i=1}^{M}\frac{m_i}{m}I_i^2\epsilon_i^2 + 100\eta^2L^2\sum_{i=1}^{M}\frac{m_i}{m}I_i^2\epsilon^2, \quad \Phi_2 = \frac{L\eta_g\eta_c\eta}{2m}\sigma^2. \tag{29}$$

This completes the proof of Theorem 1. $\qquad\qquad\square$

# D PROOF OF THEOREM 2

*Proof.* For the partial participation case, we similarly start with taking expectation over the randomness of the master round $t$ and expanding with Assumption 1 as

$$\mathbb{E}_t[f(\overline{\mathbf{x}}^{t+1})] \leq f(\overline{\mathbf{x}}^t) - \eta_g\eta_c\eta G\left\|\nabla f(\overline{\mathbf{x}}^t)\right\|^2 - \eta_g\underbrace{\mathbb{E}_t\langle\nabla f(\overline{\mathbf{x}}^t), \Delta^t - \eta_c\eta G\nabla f(\overline{\mathbf{x}}^t)\rangle}_{A_1} + \eta_g^2\frac{L}{2}\underbrace{\mathbb{E}_t\left\|\Delta^t\right\|^2}_{A_2}. \tag{30}$$

Due to Lemma 1 for both sampling strategies, $A_1$ equals exactly to the fully participating case, i.e.,

$$A_1 = \mathbb{E}_t\langle\nabla f(\overline{\mathbf{x}}^t), \Delta^t - \eta_c\eta G\nabla f(\overline{\mathbf{x}}^t)\rangle = \mathbb{E}_t\langle\nabla f(\overline{\mathbf{x}}^t), \overline{\Delta}^t - \eta_c\eta G\nabla f(\overline{\mathbf{x}}^t)\rangle. \tag{31}$$

Hence, we have exactly the same bound with the full participation case for $A_1$ as Eq. 24. Then we focus on bounding $A_2$.

Let $\theta_j^{t,\tau} = \sum_{h=0}^{I_i-1} \nabla F_j(\mathbf{x}_j^{t,\tau,h}), j \in \mathcal{V}_i$, for both sampling strategies, we have

$$
A_2 = \mathbb{E}\left\| \sum_{i=1}^{M} \frac{m_i}{m} \sum_{\tau=0}^{\omega_i-1} \eta_c \sum_{h=0}^{I_i-1} \frac{1}{n_i} \sum_{j \in \mathcal{S}_i^{t,\tau}} \eta \mathbf{g}_j^{t,\tau,h} \right\|^2
$$

$$
\overset{(a)}{=} \eta_c^2 \eta^2 \sum_{i=1}^{M} \sum_{\tau=0}^{\omega_i-1} \sum_{h=0}^{I_i-1} \sum_{j \in \mathcal{S}_i^{t,\tau}} \frac{m_i^2}{m^2} \frac{1}{n_i^2} \mathbb{E}_t \left\| \mathbf{g}_j^{t,\tau,h} - \nabla F_j(\mathbf{x}_j^{t,\tau,h}) \right\|^2 + \eta_c^2 \eta^2 \mathbb{E}_t \left\| \sum_{i=1}^{M} \frac{m_i}{mn_i} \sum_{\tau=0}^{\omega_i-1} \sum_{j \in \mathcal{S}_i^{t,\tau}} \theta_j^{t,\tau} \right\|^2
$$

$$
\overset{(b)}{\leq} G\eta_c^2 \eta^2 \sum_{i=1}^{M} \frac{m_i^2}{m^2 n_i} \sigma^2 + \eta_c^2 \eta^2 \mathbb{E}_t \left\| \sum_{i=1}^{M} \frac{m_i}{mn_i} \sum_{\tau=0}^{\omega_i-1} \sum_{j \in \mathcal{S}_i^{t,\tau}} \theta_j^{t,\tau} \right\|^2, \tag{32}
$$

where $(a)$ is due to $\mathbb{E}\|\mathbf{x}\|^2 = \mathbb{E}\|\mathbf{x} - \mathbb{E}\mathbf{x}\|^2 + \|\mathbb{E}\mathbf{x}\|^2$ and Lemma 2. $(b)$ is due to bounded local variance (Assumption 3).

We then prove a lemma that refines the second term of Eq. 32 for easier analysis.

**Lemma 6.** *For both sampling strategies, the norm of the averaged accumulated gradients of all participated workers in master round $t$ follows*

$$
\mathbb{E}_t \left\| \sum_{i=1}^{M} \frac{m_i}{mn_i} \sum_{\tau=0}^{\omega_i-1} \sum_{j \in \mathcal{S}_i^{t,\tau}} \theta_j^{t,\tau} \right\|^2
$$

$$
= \mathbb{E}_t \left[ \sum_{i=1}^{M} \frac{m_i^2}{m^2 n_i^2} \sum_{\tau=0}^{\omega_i-1} \left\| \sum_{j \in \mathcal{S}_i^{t,\tau}} \theta_j^{t,\tau} \right\|^2 - \frac{1}{m^2} \sum_{i=1}^{M} \sum_{\tau=0}^{\omega_i-1} \left\| \sum_{j \in \mathcal{V}_i} \theta_j^{t,\tau} \right\|^2 + \left\| \frac{1}{m} \sum_{i=1}^{M} \sum_{j \in \mathcal{V}_i} \sum_{\tau=0}^{\omega_i-1} \theta_j^{t,\tau} \right\|^2 \right]. \tag{33}
$$

*Proof.* We have

$$
\mathbb{E}_t \left\| \sum_{i=1}^{M} \frac{m_i}{mn_i} \sum_{\tau=0}^{\omega_i-1} \sum_{j \in \mathcal{S}_i^{t,\tau}} \theta_j^{t,\tau} \right\|^2
$$

$$
= \mathbb{E}_t \left[ \sum_{i=1}^{M} \left\| \frac{m_i}{mn_i} \sum_{\tau=0}^{\omega_i-1} \sum_{j \in \mathcal{S}_i^{t,\tau}} \theta_j^{t,\tau} \right\|^2 + \sum_{i \neq h; i,h \in [M]} \left\langle \frac{m_i}{mn_i} \sum_{\tau=0}^{\omega_i-1} \sum_{j \in \mathcal{S}_i^{t,\tau}} \theta_j^{t,\tau}, \frac{m_h}{mn_h} \sum_{\tau=0}^{\omega_h-1} \sum_{k \in \mathcal{S}_h^{t,\tau}} \theta_k^{t,\tau} \right\rangle \right], \tag{34}
$$

which is an inter-cluster decomposition among different groups. For the second term of Eq. 34, we have

$$
\mathbb{E}_t \left[ \sum_{i \neq h; i,h \in [M]} \left\langle \frac{m_i}{mn_i} \sum_{\tau=0}^{\omega_i-1} \sum_{j \in \mathcal{S}_i^{t,\tau}} \theta_j^{t,\tau}, \frac{m_h}{mn_h} \sum_{\tau=0}^{\omega_h-1} \sum_{k \in \mathcal{S}_h^{t,\tau}} \theta_k^{t,\tau} \right\rangle \right]
$$

$$
= \mathbb{E}_t \left[ \sum_{i \neq h; i,h \in [M]} \frac{m_i m_h}{m^2} \left\langle \frac{1}{n_i} \sum_{\tau=0}^{\omega_i-1} \sum_{j \in \mathcal{S}_i^{t,\tau}} \theta_j^{t,\tau}, \frac{1}{n_h} \sum_{\tau=0}^{\omega_h-1} \sum_{k \in \mathcal{S}_h^{t,\tau}} \theta_k^{t,\tau} \right\rangle \right]
$$

$$
\overset{(a)}{=} \frac{1}{m^2} \mathbb{E}_t \left[ \sum_{i \neq h; i,h \in [M]} \left\langle \sum_{j \in \mathcal{V}_i} \sum_{\tau=0}^{\omega_i-1} \theta_j^{t,\tau}, \sum_{k \in \mathcal{V}_h} \sum_{\tau=0}^{\omega_h-1} \theta_k^{t,\tau} \right\rangle \right], \tag{35}
$$

where $(a)$ is due to Lemma 1 and the independence among the sampling sets of different groups.

We note that it is a fact that

$$
\mathbb{E}_t \left\| \frac{1}{m} \sum_{i=1}^{M} \sum_{j \in \mathcal{V}_i} \sum_{\tau=0}^{\omega_i-1} \theta_j^{t,\tau} \right\|^2
$$

$$
= \frac{1}{m^2} \mathbb{E}_t \left[ \sum_{i=1}^{M} \left\| \sum_{j \in \mathcal{V}_i} \sum_{\tau=0}^{\omega_i-1} \theta_j^{t,\tau} \right\|^2 + \sum_{i \neq h, i,h \in [M]} \left\langle \sum_{j \in \mathcal{V}_i} \sum_{\tau=0}^{\omega_i-1} \theta_j^{t,\tau}, \sum_{k \in \mathcal{V}_h} \sum_{\tau=0}^{\omega_h-1} \theta_k^{t,\tau} \right\rangle \right]. \tag{36}
$$

Then, review the first term of Eq. 34, we have

$$\mathbb{E}_t \left\| \sum_{\tau=0}^{\omega_i-1} \sum_{j \in \mathcal{S}_i^{t,\tau}} \theta_j^{t,\tau} \right\|^2 = \mathbb{E}_t \left[ \sum_{\tau=0}^{\omega_i-1} \left\| \sum_{j \in \mathcal{S}_i^{t,\tau}} \theta_j^{t,\tau} \right\|^2 + \sum_{\tau \neq \nu; \tau, \nu \in [\omega_i-1]} \left\langle \sum_{j \in \mathcal{S}_i^{t,\tau}} \theta_j^{t,\tau}, \sum_{k \in \mathcal{S}_i^{t,\nu}} \theta_k^{t,\nu} \right\rangle \right], \tag{37}$$

which is an intra-cluster decomposition among different cluster rounds. Note that for the second term of Eq. 37, we have

$$\mathbb{E}_t \left[ \sum_{\tau \neq \nu; \tau, \nu \in [\omega_i-1]} \left\langle \sum_{j \in \mathcal{S}_i^{t,\tau}} \theta_j^{t,\tau}, \sum_{k \in \mathcal{S}_i^{t,\nu}} \theta_k^{t,\nu} \right\rangle \right]$$

$$= \mathbb{E}_t \left[ \sum_{\tau \neq \nu; \tau, \nu \in [\omega_i-1]} n_i^2 \left\langle \frac{1}{n_i} \sum_{j \in \mathcal{S}_i^{t,\tau}} \theta_j^{t,\tau}, \frac{1}{n_i} \sum_{k \in \mathcal{S}_i^{t,\nu}} \theta_k^{t,\nu} \right\rangle \right]$$

$$\overset{(a)}{=} \frac{n_i^2}{m_i^2} \mathbb{E}_t \left[ \sum_{\tau \neq \nu; \tau, \nu \in [\omega_i-1]} \left\langle \sum_{j \in \mathcal{V}_i} \theta_j^{t,\tau}, \sum_{k \in \mathcal{V}_i} \theta_k^{t,\nu} \right\rangle \right], \tag{38}$$

where $(a)$ is due to Lemma 1 and the independence among the sampling sets of different cluster rounds.

We similarly note that it is fact for the first term of Eq. 36 that

$$\mathbb{E}_t \left\| \sum_{j \in \mathcal{V}_i} \sum_{\tau=0}^{\omega_i-1} \theta_j^{t,\tau} \right\|^2 = \mathbb{E}_t \left[ \sum_{\tau=0}^{\omega_i-1} \left\| \sum_{j \in \mathcal{V}_i} \theta_j^{t,\tau} \right\|^2 + \sum_{\tau \neq \nu; \tau, \nu \in [\omega_i-1]} \left\langle \sum_{j \in \mathcal{V}_i} \theta_j^{t,\tau}, \sum_{k \in \mathcal{V}_i} \theta_k^{t,\nu} \right\rangle \right]. \tag{39}$$

Hence, substituting all Eq. 35, 36, 37, 38, and 39 back into Eq. 34, we have

$$\mathbb{E}_t \left\| \sum_{i=1}^{M} \frac{m_i}{mn_i} \sum_{\tau=0}^{\omega_i-1} \sum_{j \in \mathcal{S}_i^{t,\tau}} \theta_j^{t,\tau} \right\|^2$$

$$= \mathbb{E}_t \left[ \sum_{i=1}^{M} \frac{m_i^2}{m^2 n_i^2} \sum_{\tau=0}^{\omega_i-1} \left\| \sum_{j \in \mathcal{S}_i^{t,\tau}} \theta_j^{t,\tau} \right\|^2 + \frac{1}{m^2} \sum_{i=1}^{M} \left( \left\| \sum_{j \in \mathcal{V}_i} \sum_{\tau=0}^{\omega_i-1} \theta_j^{t,\tau} \right\|^2 - \sum_{\tau=0}^{\omega_i-1} \left\| \sum_{j \in \mathcal{V}_i} \theta_j^{t,\tau} \right\|^2 \right) \right.$$

$$+ \left. \left\| \frac{1}{m} \sum_{i=1}^{M} \sum_{j \in \mathcal{V}_i} \sum_{\tau=0}^{\omega_i-1} \theta_j^{t,\tau} \right\|^2 - \frac{1}{m^2} \sum_{i=1}^{M} \left\| \sum_{j \in \mathcal{V}_i} \sum_{\tau=0}^{\omega_i-1} \theta_j^{t,\tau} \right\|^2 \right]$$

$$= \mathbb{E}_t \left[ \sum_{i=1}^{M} \frac{m_i^2}{m^2 n_i^2} \sum_{\tau=0}^{\omega_i-1} \left\| \sum_{j \in \mathcal{S}_i^{t,\tau}} \theta_j^{t,\tau} \right\|^2 - \frac{1}{m^2} \sum_{i=1}^{M} \sum_{\tau=0}^{\omega_i-1} \left\| \sum_{j \in \mathcal{V}_i} \theta_j^{t,\tau} \right\|^2 + \left\| \frac{1}{m} \sum_{i=1}^{M} \sum_{j \in \mathcal{V}_i} \sum_{\tau=0}^{\omega_i-1} \theta_j^{t,\tau} \right\|^2 \right]. \tag{40}$$

This completes the proof of Lemma 6. $\qquad\square$

Note that Lemma 6 essentially redirect the uncertainty of partial participation from overall master round level to each cluster round level (the first term of Eq. 40). It is this redirection that restricts partial participation to only directly interact with $I_i$ rather than $G$, thereby guaranteeing the weakening effect.

With Lemma 6, we have $A_2$ for both sampling strategies

$$A_2 \leq G \eta_c^2 \eta^2 \sum_{i=1}^{M} \frac{m_i^2}{m^2 n_i} \sigma^2 + \eta_c^2 \eta^2 \sum_{i=1}^{M} \frac{m_i^2}{m^2 n_i^2} \sum_{\tau=0}^{\omega_i-1} \mathbb{E}_t \left\| \sum_{j \in \mathcal{S}_i^{t,\tau}} \theta_j^{t,\tau} \right\|^2 \tag{41}$$

$$- \eta_c^2 \eta^2 \frac{1}{m^2} \sum_{i=1}^{M} \sum_{\tau=0}^{\omega_i-1} \mathbb{E}_t \left\| \sum_{j \in \mathcal{V}_i} \theta_j^{t,\tau} \right\|^2 + \eta_c^2 \eta^2 \mathbb{E}_t \left\| \frac{1}{m} \sum_{i=1}^{M} \sum_{j \in \mathcal{V}_i} \sum_{\tau=0}^{\omega_i-1} \theta_j^{t,\tau} \right\|^2.$$

Note that only the second term of Eq. 41 remains relevant to specific sampling strategies. We will next bound for it.

## D.1 BOUNDING STRATEGY 1

With sampling strategy 1, suppose $\mathcal{S}_i^{t,\tau} = \{l_{i,1}^{t,\tau}, l_{i,2}^{t,\tau}, \ldots, l_{i,n_i}^{t,\tau}\}$, we have for the second term of Eq. 41

$$
\mathbb{E}_t \left\| \sum_{j \in \mathcal{S}_i^{t,\tau}} \theta_j^{t,\tau} \right\|^2 = \mathbb{E}_t \left\| \sum_{z=1}^{n_i} \theta_{l_{i,z}^{t,\tau}}^{t,\tau} \right\|^2
$$

$$
= \mathbb{E}_t \left[ \sum_{z=1}^{n_i} \left\| \theta_{l_{i,z}^{t,\tau}}^{t,\tau} \right\|^2 + \sum_{j \neq k; j,k \in [n_i]} \left\langle \theta_{l_{i,j}^{t,\tau}}^{t,\tau}, \theta_{l_{i,k}^{t,\tau}}^{t,\tau} \right\rangle \right]
$$

$$
= \mathbb{E}_t \left[ n_i \left\| \theta_{l_{i,1}^{t,\tau}}^{t,\tau} \right\|^2 + n_i(n_i - 1) \left\langle \theta_{l_{i,1}^{t,\tau}}^{t,\tau}, \theta_{l_{i,2}^{t,\tau}}^{t,\tau} \right\rangle \right]
$$

$$
= \mathbb{E}_t \left[ \sum_{j \in \mathcal{V}_i} \frac{n_i}{m_i} \left\| \theta_j^{t,\tau} \right\|^2 + \sum_{j,k \in \mathcal{V}_i} \frac{n_i(n_i - 1)}{m_i^2} \left\langle \theta_j^{t,\tau}, \theta_k^{t,\tau} \right\rangle \right]
$$

$$
= \sum_{j \in \mathcal{V}_i} \frac{n_i}{m_i} \mathbb{E}_t \left\| \theta_j^{t,\tau} \right\|^2 + \frac{n_i(n_i - 1)}{m_i^2} \mathbb{E}_t \left\| \sum_{j \in \mathcal{V}_i} \theta_j^{t,\tau} \right\|^2 . \tag{42}
$$

Substituting Eq. 42 back into Eq. 41, we have

$$
A_2 \leq G\eta_c^2 \eta^2 \sum_{i=1}^{M} \frac{m_i^2}{m^2 n_i} \sigma^2 + \eta_c^2 \eta^2 \sum_{i=1}^{M} \sum_{\tau=0}^{\omega_i - 1} \sum_{j \in \mathcal{V}_i} \frac{m_i}{m^2 n_i} \mathbb{E}_t \left\| \theta_j^{t,\tau} \right\|^2
$$

$$
- \eta_c^2 \eta^2 \frac{1}{m^2} \sum_{i=1}^{M} \frac{1}{n_i} \sum_{\tau=0}^{\omega_i - 1} \mathbb{E}_t \left\| \sum_{j \in \mathcal{V}_i} \theta_j^{t,\tau} \right\|^2 + \eta_c^2 \eta^2 \mathbb{E}_t \left\| \frac{1}{m} \sum_{i=1}^{M} \sum_{j \in \mathcal{V}_i} \sum_{\tau=0}^{\omega_i - 1} \theta_j^{t,\tau} \right\|^2
$$

$$
\leq G\eta_c^2 \eta^2 \sum_{i=1}^{M} \frac{m_i^2}{m^2 n_i} \sigma^2 + \eta_c^2 \eta^2 \sum_{i=1}^{M} \sum_{\tau=0}^{\omega_i - 1} \sum_{j \in \mathcal{V}_i} \frac{m_i}{m^2 n_i} \mathbb{E}_t \left\| \theta_j^{t,\tau} \right\|^2 + \eta_c^2 \eta^2 \mathbb{E}_t \left\| \frac{1}{m} \sum_{i=1}^{M} \sum_{j \in \mathcal{V}_i} \sum_{\tau=0}^{\omega_i - 1} \theta_j^{t,\tau} \right\|^2 \tag{43}
$$

For the second term of Eq. 43, we can have the following important inequality. Here we similarly consider arbitrary positive coefficients $p_i > 0, \forall i \in [M]$ for compatibility with our subsequent proof for strategy 2,

$$
\sum_{i=1}^{M} p_i \sum_{\tau=0}^{\omega_i - 1} \sum_{j \in \mathcal{V}_i} \mathbb{E}_t \left\| \sum_{h=0}^{I_i - 1} \nabla F_j(\mathbf{x}_j^{t,\tau,h}) \right\|^2 = \sum_{i=1}^{M} p_i \sum_{\tau=0}^{\omega_i - 1} \sum_{j \in \mathcal{V}_i} \mathbb{E}_t \left\| \sum_{h=0}^{I_i - 1} \left( \nabla F_j(\mathbf{x}_j^{t,\tau,h}) - \nabla F_j(\bar{\mathbf{x}}_i^{t,\tau}) \right. \right.
$$

$$
\left. \left. + \nabla F_j(\bar{\mathbf{x}}_i^{t,\tau}) - \nabla f_i(\bar{\mathbf{x}}_i^{t,\tau}) + \nabla f_i(\bar{\mathbf{x}}_i^{t,\tau}) - \nabla f_i(\bar{\mathbf{x}}^t) + \nabla f_i(\bar{\mathbf{x}}^t) - \nabla f(\bar{\mathbf{x}}^t) + \nabla f(\bar{\mathbf{x}}^t) \right) \right\|^2
$$

$$
\leq 5L^2 \sum_{i=1}^{M} p_i I_i \sum_{\tau=0}^{\omega_i - 1} \sum_{j \in \mathcal{V}_i} \sum_{h=0}^{I_i - 1} \mathbb{E}_t \left\| \mathbf{x}_j^{t,\tau,h} - \bar{\mathbf{x}}_i^{t,\tau} \right\|^2 + 5L^2 \sum_{i=1}^{M} p_i m_i I_i^2 \sum_{\tau=0}^{\omega_i - 1} \mathbb{E}_t \left\| \mathbf{x}_i^{t,\tau} - \bar{\mathbf{x}}^t \right\|^2
$$

$$
+ 5G \sum_{i=1}^{M} p_i m_i I_i \epsilon_i^2 + 5G \sum_{i=1}^{M} p_i m_i I_i \epsilon^2 + 5G \sum_{i=1}^{M} p_i m_i I_i \| \nabla f(\bar{\mathbf{x}}^t) \|^2
$$

$$
\stackrel{(a)}{\leq} \frac{31}{5} L^2 \sum_{i=1}^{M} p_i I_i \sum_{\tau=0}^{\omega_i - 1} \sum_{j \in \mathcal{V}_i} \sum_{h=0}^{I_i - 1} \mathbb{E}_t \left\| \mathbf{x}_j^{t,\tau,h} - \bar{\mathbf{x}}_i^{t,\tau} \right\|^2 + 5G \sum_{i=1}^{M} p_i m_i I_i \epsilon_i^2 + 7G \sum_{i=1}^{M} p_i m_i I_i \epsilon^2
$$

$$
+ 7G \sum_{i=1}^{M} p_i m_i I_i \| \nabla f(\bar{\mathbf{x}}^t) \|^2 + \frac{1}{4} \sum_{i=1}^{M} \frac{p_i m_i}{n_i} I_i \sigma^2 + 15G\eta_c^2 \eta^2 L^2 \sum_{i=1}^{M} p_i m_i I_i \sum_{\tau=0}^{\omega_i - 1} \Psi_i^{t,\tau}
$$

$$
\stackrel{(b)}{\leq} \frac{2}{5} G \sum_{i=1}^{M} p_i m_i \sigma^2 + \frac{1}{2} \sum_{i=1}^{M} \frac{p_i m_i}{n_i} I_i \sigma^2 + 9G \sum_{i=1}^{M} p_i m_i I_i \epsilon_i^2 + 12G \sum_{i=1}^{M} p_i m_i I_i \epsilon^2
$$

$$
+ 12G \sum_{i=1}^{M} p_i m_i I_i \| \nabla f(\bar{\mathbf{x}}^t) \|^2 + 28G\eta_c^2 \eta^2 L^2 \sum_{i=1}^{M} p_i m_i I_i \sum_{\tau=0}^{\omega_i - 1} \Psi_i^{t,\tau} , \tag{44}
$$

where $(a)$ is due to Lemma 3 for partial participation and simplified with $\eta_c\eta \leq \frac{1}{10LG}$. $(b)$ is due to Lemma 5 for partial participation case and simplified via the condition $\eta \leq \frac{1}{10I_{max}L}$.

We can bound $\Psi_i^{t,\tau}$ as

$$
\begin{aligned}
\Psi_i^{t,\tau} &= \mathbb{E}_t\left\|\frac{1}{n_i}\sum_{j\in\mathcal{S}_i^{t,\tau}}\sum_{h=0}^{I_i}\nabla F_j(\mathbf{x}_j^{t,\tau,h}) - \frac{1}{m_i}\sum_{j\in\mathcal{V}_i}\sum_{h=0}^{I_i}\nabla F_j(\mathbf{x}_j^{t,\tau,h})\right\|^2 \\
&\stackrel{(a)}{=} \mathbb{E}_t\left\|\frac{1}{n_i}\sum_{j\in\mathcal{S}_i^{t,\tau}}\sum_{h=0}^{I_i}\nabla F_j(\mathbf{x}_j^{t,\tau,h})\right\|^2 - \mathbb{E}_t\left\|\frac{1}{m_i}\sum_{j\in\mathcal{V}_i}\sum_{h=0}^{I_i}\nabla F_j(\mathbf{x}_j^{t,\tau,h})\right\|^2 \\
&= \frac{1}{n_i^2}\mathbb{E}_t\left\|\sum_{j\in\mathcal{S}_i^{t,\tau}}\theta_j^{t,\tau}\right\|^2 - \frac{1}{m_i^2}\mathbb{E}_t\left\|\sum_{j\in\mathcal{V}_i}\theta_j^{t,\tau}\right\|^2,
\end{aligned}
\tag{45}
$$

where $(a)$ is due to the fact that $\mathbb{E}\|\mathbf{x}\|^2 = \mathbb{E}\|\mathbf{x}-\mathbb{E}\mathbf{x}\|^2 + \|\mathbb{E}\mathbf{x}\|^2$, holding for both sampling strategies.

For sampling strategy 1, substituting Eq. 42 into Eq. 45, we have

$$
\Psi_i^{t,\tau} = \frac{1}{n_im_i}\sum_{j\in\mathcal{V}_i}\mathbb{E}_t\|\theta_j^{t,\tau}\|^2 - \frac{1}{n_im_i^2}\mathbb{E}_t\left\|\sum_{j\in\mathcal{V}_i}\theta_j^{t,\tau}\right\|^2,
\tag{46}
$$

Substituting Eq. 46 back into Eq. 44 with $p_i = \frac{m_i}{m^2 n_i}, \forall i \in [M]$, we have

$$
\begin{aligned}
&\sum_{i=1}^{M}\frac{m_i}{m^2 n_i}\sum_{\tau=0}^{\omega_i-1}\sum_{j\in\mathcal{V}_i}\mathbb{E}_t\|\theta_j^{t,\tau}\|^2 \\
&\leq \frac{2}{5}G\sum_{i=1}^{M}\frac{m_i^2}{m^2 n_i}\sigma^2 + \frac{1}{2}\sum_{i=1}^{M}\frac{m_i^2}{m^2 n_i^2}I_i\sigma^2 + 9G\sum_{i=1}^{M}\frac{m_i^2}{m^2 n_i}I_i\epsilon_i^2 \\
&\quad + 12G\sum_{i=1}^{M}\frac{m_i^2}{m^2 n_i}I_i\epsilon^2 + 12G\sum_{i=1}^{M}\frac{m_i^2}{m^2 n_i}I_i\|\nabla f(\overline{\mathbf{x}}^t)\|^2 \\
&\quad + 28G\eta_c^2\eta^2 L^2\sum_{i=1}^{M}\frac{m_i}{m^2 n_i^2}I_i\sum_{\tau=0}^{\omega_i-1}\sum_{j\in\mathcal{V}_i}\mathbb{E}_t\|\theta_j^{t,\tau}\|^2 - 28G\eta_c^2\eta^2 L^2\sum_{i=1}^{M}\frac{1}{m^2 n_i^2}I_i\sum_{\tau=0}^{\omega_i-1}\mathbb{E}_t\left\|\sum_{j\in\mathcal{V}_i}\theta_j^{t,\tau}\right\|^2 \\
&\stackrel{(a)}{\leq} \frac{9}{10}G\sum_{i=1}^{M}\frac{m_i^2}{m^2 n_i}\sigma^2 + 9G\sum_{i=1}^{M}\frac{m_i^2}{m^2 n_i}I_i\epsilon_i^2 + 12G\sum_{i=1}^{M}\frac{m_i^2}{m^2 n_i}I_i\epsilon^2 \\
&\quad + 12G\sum_{i=1}^{M}\frac{m_i^2}{m^2 n_i}I_i\|\nabla f(\overline{\mathbf{x}}^t)\|^2 + \frac{2}{5}\sum_{i=1}^{M}\frac{m_i}{m^2 n_i}\sum_{\tau=0}^{\omega_i-1}\sum_{j\in\mathcal{V}_i}\mathbb{E}_t\|\theta_j^{t,\tau}\|^2,
\end{aligned}
\tag{47}
$$

where $(a)$ is due to the fact that $\frac{I_i}{n_i} < G, \forall i \in [M]$ and the condition $\eta_c\eta \leq \frac{1}{10LG}$.

Rearranging the order for the left hand term, we have

$$
\begin{aligned}
&\sum_{i=1}^{M}\frac{m_i}{m^2 n_i}\sum_{\tau=0}^{\omega_i-1}\sum_{j\in\mathcal{V}_i}\mathbb{E}_t\|\theta_j^{t,\tau}\|^2 \\
&\leq \frac{3}{2}G\sum_{i=1}^{M}\frac{m_i^2}{m^2 n_i}\sigma^2 + 15G\sum_{i=1}^{M}\frac{m_i^2}{m^2 n_i}I_i\epsilon_i^2 + 20G\sum_{i=1}^{M}\frac{m_i^2}{m^2 n_i}I_i\epsilon^2 + 20G\sum_{i=1}^{M}\frac{m_i^2}{m^2 n_i}I_i\|\nabla f(\overline{\mathbf{x}}^t)\|^2.
\end{aligned}
\tag{48}
$$

We substitute Eq. 48 back to Eq. 43, and bound $A_2$ for strategy 1 as

$$
A_2 \leq \frac{5}{2}G\eta_c^2\eta^2\sum_{i=1}^{M}\frac{m_i^2}{m^2 n_i}\sigma^2 + \eta_c^2\eta^2\mathbb{E}_t\left\|\frac{1}{m}\sum_{i=1}^{M}\sum_{j\in\mathcal{V}_i}\sum_{\tau=0}^{\omega_i-1}\sum_{h=0}^{I_i-1}\nabla F_j(\mathbf{x}_j^{t,\tau,h})\right\|^2
$$

$$+ 15G\eta_c^2\eta^2 \sum_{i=1}^{M} \frac{m_i^2}{m^2 n_i} I_i \epsilon_i^2 + 20G\eta_c^2\eta^2 \sum_{i=1}^{M} \frac{m_i^2}{m^2 n_i} I_i \epsilon^2 + 20G\eta_c^2\eta^2 \sum_{i=1}^{M} \frac{m_i^2}{m^2 n_i} I_i \|\nabla f(\bar{\mathbf{x}}^t)\|^2. \tag{49}$$

Substituting Eq. 24 and Eq. 49 back into Eq. 30 and rearranging the order, we have

$$\eta_g\eta_c\eta G\left(\frac{1}{2} - 40G^2\eta_c^2\eta^2 L^2 - 100\eta^2 L^2 \sum_{i=1}^{M} \frac{m_i}{m} I_i^2 - 10\eta_g\eta_c\eta L \sum_{i=1}^{M} \frac{m_i^2}{m^2 n_i} I_i\right) \left\|\nabla f(\bar{\mathbf{x}}^t)\right\|^2$$

$$\leq f(\bar{\mathbf{x}}^t) - \mathbb{E}_t[f(\bar{\mathbf{x}}^{t+1})] + 9G^2\eta_g\eta_c^3\eta^3 L^2 \frac{M}{m}\sigma^2 + 8G\eta_g\eta_c\eta^3 L^2 \sum_{i=1}^{M} \frac{m_i}{m} I_i \sigma^2$$

$$+ 75G\eta_g\eta_c\eta^3 L^2 \sum_{i=1}^{M} \frac{m_i}{m} I_i^2 \epsilon_i^2 + 100G\eta_g\eta_c\eta^3 L^2 \sum_{i=1}^{M} \frac{m_i}{m} I_i^2 \epsilon^2 + 40G^3\eta_g\eta_c^3\eta^3 L^2 \epsilon^2$$

$$+ \frac{5}{4}G\eta_g^2\eta_c^2\eta^2 L \sum_{i=1}^{M} \frac{m_i^2}{m^2 n_i}\sigma^2 + \frac{15}{2}G\eta_g^2\eta_c^2\eta^2 L \sum_{i=1}^{M} \frac{m_i^2}{m^2 n_i} I_i \epsilon_i^2 + 10G\eta_g^2\eta_c^2\eta^2 L \sum_{i=1}^{M} \frac{m_i^2}{m^2 n_i} I_i \epsilon^2. \tag{50}$$

Here, like we do in the full case, we drop the term $\mathbb{E}_t\|\frac{1}{m}\sum_{i=1}^{M}\sum_{\tau=0}^{\omega_i-1}\sum_{h=0}^{I_i-1}\sum_{j\in\mathcal{V}_i}\nabla F_j(\mathbf{x}_j^{t,\tau,h})\|^2$ with condition $\eta_g\eta_c\eta \leq \frac{1}{GL}$ ensuring its coefficient $\frac{L\eta_g^2\eta_c^2\eta^2}{2} - \frac{\eta_g\eta_c\eta}{2G} \leq 0$.

Suppose $40G^2\eta_c^2\eta^2 L^2 + 100\eta^2 L^2 \sum_{i=1}^{M} \frac{m_i}{m} I_i^2 + 10\eta_g\eta_c\eta L \sum_{i=1}^{M} \frac{m_i^2}{m^2 n_i} I_i < \frac{1}{2}$, and there exists a constant $c > 0$ satisfying $(\frac{1}{2} - 40G^2\eta_c^2\eta^2 L^2 - 100\eta^2 L^2 \sum_{i=1}^{M} \frac{m_i}{m} I_i^2 - 10\eta_g\eta_c\eta L \sum_{i=1}^{M} \frac{m_i^2}{m^2 n_i} I_i) > c > 0$, then we have

$$\left\|\nabla f(\bar{\mathbf{x}}^t)\right\|^2 \leq \frac{f(\bar{\mathbf{x}}^t) - \mathbb{E}_t[f(\bar{\mathbf{x}}^{t+1})]}{\eta_g\eta_c\eta Gc} + \frac{1}{c}\Bigg[9G\eta_c^2\eta^2 L^2 \frac{M}{m}\sigma^2 + 8\eta^2 L^2 \sum_{i=1}^{M} \frac{m_i}{m} I_i \sigma^2$$

$$+ 75\eta^2 L^2 \sum_{i=1}^{M} \frac{m_i}{m} I_i^2 \epsilon_i^2 + 100\eta^2 L^2 \sum_{i=1}^{M} \frac{m_i}{m} I_i^2 \epsilon^2 + 40G^2\eta_c^2\eta^2 L^2 \epsilon^2$$

$$+ \frac{5}{4}\eta_g\eta_c\eta L \sum_{i=1}^{M} \frac{m_i^2}{m^2 n_i}\sigma^2 + \frac{15}{2}\eta_g\eta_c\eta L \sum_{i=1}^{M} \frac{m_i^2}{m^2 n_i} I_i \epsilon_i^2 + 10\eta_g\eta_c\eta L \sum_{i=1}^{M} \frac{m_i^2}{m^2 n_i} I_i \epsilon^2\Bigg], \tag{51}$$

Taking a double expectation over the data samples among all workers and averaging from $t = 0, 1, \ldots, T$, we have the final results as

$$\min_{t\in[T]} \mathbb{E}\left\|\nabla f(\bar{\mathbf{x}}^t)\right\|^2 \leq \frac{f_0 - f_*}{c\eta_g\eta_c\eta GT} + \frac{1}{c}(\Phi_1 + \Phi_2 + \Phi_3), \tag{52}$$

where,

$$\Phi_1 = 9G\eta_c^2\eta^2 L^2 \frac{M}{m}\sigma^2 + 8\eta^2 L^2 \sum_{i=1}^{M} \frac{m_i}{m} I_i \sigma^2 + 40G^2\eta_c^2\eta^2 L^2 \epsilon^2$$

$$+ +75\eta^2 L^2 \sum_{i=1}^{M} \frac{m_i}{m} I_i^2 \epsilon_i^2 + 100\eta^2 L^2 \sum_{i=1}^{M} \frac{m_i}{m} I_i^2 \epsilon^2, \quad \Phi_2 = \frac{1}{2}\eta_g\eta_c\eta L \sum_{i=1}^{M} \frac{m_i^2}{m^2 n_i}\sigma^2,$$

$$\Phi_3 = \frac{3}{4}\eta_g\eta_c\eta L \sum_{i=1}^{M} \frac{m_i^2}{m^2 n_i}\sigma^2 + \frac{15}{2}\eta_g\eta_c\eta L \sum_{i=1}^{M} \frac{m_i^2}{m^2 n_i} I_i \epsilon_i^2 + 10\eta_g\eta_c\eta L \sum_{i=1}^{M} \frac{m_i^2}{m^2 n_i} I_i \epsilon^2. \tag{53}$$

### D.2 BOUNDING STRATEGY 2

With sampling strategy 2, we have for the second term of Eq. 41

$$\mathbb{E}_t\Bigg\|\sum_{j\in\mathcal{S}_i^{t,\tau}} \theta_j^{t,\tau}\Bigg\|^2 = \mathbb{E}_t\Bigg\|\sum_{j\in\mathcal{V}_i} \mathbb{P}\{j\in\mathcal{S}_i^{t,\tau}\}\theta_j^{t,\tau}\Bigg\|^2$$

$$= \mathbb{E}_t\left[\sum_{j\in\mathcal{V}_i}\mathbb{P}\{j\in\mathcal{S}_i^{t,\tau}\}\|\theta_j^{t,\tau}\|^2 + \sum_{j\neq k;j,k\in\mathcal{V}_i}\mathbb{P}\{j,k\in\mathcal{S}_i^{t,\tau}\}\langle\theta_j^{t,\tau},\theta_k^{t,\tau}\rangle\right]$$

$$= \mathbb{E}_t\left[\frac{n_i}{m_i}\sum_{j\in\mathcal{V}_i}\|\theta_j^{t,\tau}\|^2 + \frac{n_i(n_i-1)}{m_i(m_i-1)}\sum_{j\neq k;j,k\in\mathcal{V}_i}\langle\theta_j^{t,\tau},\theta_k^{t,\tau}\rangle\right]$$

$$= \mathbb{E}_t\left[\frac{n_i}{m_i}\sum_{j\in\mathcal{V}_i}\|\theta_j^{t,\tau}\|^2 + \frac{n_i(n_i-1)}{m_i(m_i-1)}\left(\Big\|\sum_{j\in\mathcal{V}_i}\theta_j^{t,\tau}\Big\|^2 - \sum_{j\in\mathcal{V}_i}\|\theta_j^{t,\tau}\|^2\right)\right]$$

$$= \sum_{j\in\mathcal{V}_i}\frac{n_i(m_i-n_i)}{m_i(m_i-1)}\mathbb{E}_t\|\theta_j^{t,\tau}\|^2 + \frac{n_i(n_i-1)}{m_i(m_i-1)}\mathbb{E}_t\Big\|\sum_{j\in\mathcal{V}_i}\theta_j^{t,\tau}\Big\|^2. \tag{54}$$

Substituting Eq. 54 back into Eq. 41, we have

$$A_2 \leq G\eta_c^2\eta^2\sum_{i=1}^{M}\frac{m_i^2}{m^2 n_i}\sigma^2 + \eta_c^2\eta^2\sum_{i=1}^{M}\sum_{\tau=0}^{\omega_i-1}\sum_{j\in\mathcal{V}_i}\frac{m_i(m_i-n_i)}{m^2 n_i(m_i-1)}\mathbb{E}_t\|\theta_j^{t,\tau}\|^2$$

$$- \eta_c^2\eta^2\frac{1}{m^2}\sum_{i=1}^{M}\frac{m_i-n_i}{n_i(m_i-1)}\sum_{\tau=0}^{\omega_i-1}\mathbb{E}_t\Big\|\sum_{j\in\mathcal{V}_i}\theta_j^{t,\tau}\Big\|^2 + \eta_c^2\eta^2\mathbb{E}_t\Big\|\frac{1}{m}\sum_{i=1}^{M}\sum_{j\in\mathcal{V}_i}\sum_{\tau=0}^{\omega_i-1}\theta_j^{t,\tau}\Big\|^2$$

$$\leq G\eta_c^2\eta^2\sum_{i=1}^{M}\frac{m_i^2}{m^2 n_i}\sigma^2 + \eta_c^2\eta^2\sum_{i=1}^{M}\sum_{\tau=0}^{\omega_i-1}\sum_{j\in\mathcal{V}_i}\frac{m_i(m_i-n_i)}{m^2 n_i(m_i-1)}\mathbb{E}_t\|\theta_j^{t,\tau}\|^2 + \eta_c^2\eta^2\mathbb{E}_t\Big\|\frac{1}{m}\sum_{i=1}^{M}\sum_{j\in\mathcal{V}_i}\sum_{\tau=0}^{\omega_i-1}\theta_j^{t,\tau}\Big\|^2 \tag{55}$$

Let $\alpha_i = \frac{m_i-n_i}{m_i-1}$. Like we do in strategy 1, we can also bound $\Psi_i^{t,\tau}$ for strategy 2 by substituting Eq. 54 into the first term of Eq. 45

$$\Psi_i^{t,\tau} = \sum_{j\in\mathcal{V}_i}\frac{\alpha_i}{m_i n_i}\mathbb{E}_t\|\theta_j^{t,\tau}\|^2 - \frac{\alpha_i}{m_i^2 n_i}\mathbb{E}_t\Big\|\sum_{j\in\mathcal{V}_i}\theta_j^{t,\tau}\Big\|^2 \tag{56}$$

Similarly, substituting Eq. 56 back into Eq. 44 with $p_i = \frac{m_i\alpha_i}{m^2 n_i}, \forall i\in[M]$, we have

$$\sum_{i=1}^{M}\frac{m_i\alpha_i}{m^2 n_i}\sum_{\tau=0}^{\omega_i-1}\sum_{j\in\mathcal{V}_i}\mathbb{E}_t\Big\|\sum_{h=0}^{I_i-1}\nabla F_j(\mathbf{x}_j^{t,\tau,h})\Big\|^2$$

$$\leq \frac{9}{10}G\sum_{i=1}^{M}\frac{m_i^2\alpha_i}{m^2 n_i}\sigma^2 + 9G\sum_{i=1}^{M}\frac{m_i^2\alpha_i}{m^2 n_i}I_i\epsilon_i^2 + 12G\sum_{i=1}^{M}\frac{m_i^2\alpha_i}{m^2 n_i}I_i\epsilon^2 + 12G\sum_{i=1}^{M}\frac{m_i^2\alpha_i}{m^2 n_i}I_i\|\nabla f(\bar{\mathbf{x}}^t)\|^2$$

$$+ \frac{2}{5}\sum_{i=1}^{M}\frac{m_i\alpha_i}{m^2 n_i}\sum_{\tau=0}^{\omega_i-1}\sum_{j\in\mathcal{V}_i}\mathbb{E}_t\|\theta_j^{t,\tau}\|^2, \tag{57}$$

where $(a)$ is due to the fact that $\frac{I_i}{n_i} < G$ and $\alpha_i \leq 1, \forall i\in[M]$ and the condition $\eta_c\eta \leq \frac{1}{10LG}$.

Rearranging the order for the left hand term, we have

$$\sum_{i=1}^{M}\frac{m_i\alpha_i}{m^2 n_i}\sum_{\tau=0}^{\omega_i-1}\sum_{j\in\mathcal{V}_i}\mathbb{E}_t\|\theta_j^{t,\tau}\|^2$$

$$\leq \frac{3}{2}G\sum_{i=1}^{M}\frac{m_i^2\alpha_i}{m^2 n_i}\sigma^2 + 15G\sum_{i=1}^{M}\frac{m_i^2\alpha_i}{m^2 n_i}I_i\epsilon_i^2 + 20G\sum_{i=1}^{M}\frac{m_i^2\alpha_i}{m^2 n_i}I_i\epsilon^2 + 20G\sum_{i=1}^{M}\frac{m_i^2\alpha_i}{m^2 n_i}I_i\|\nabla f(\bar{\mathbf{x}}^t)\|^2. \tag{58}$$

We substitute Eq. 58 back to Eq. 55, and bound $A_2$ for strategy 2 as

$$A_2 \leq G\eta_c^2\eta^2\sum_{i=1}^{M}\frac{m_i^2}{m^2 n_i}\sigma^2 + \eta_c^2\eta^2\mathbb{E}_t\Big\|\frac{1}{m}\sum_{i=1}^{M}\sum_{j\in\mathcal{V}_i}\sum_{\tau=0}^{\omega_i-1}\sum_{h=0}^{I_i-1}\nabla F_j(\mathbf{x}_j^{t,\tau,h})\Big\|^2 + \frac{3}{2}G\eta_c^2\eta^2\sum_{i=1}^{M}\frac{m_i^2\alpha_i}{m^2 n_i}\sigma^2$$

$$+ 15G\eta_c^2\eta^2 \sum_{i=1}^{M} \frac{m_i^2\alpha_i}{m^2 n_i} I_i\epsilon_i^2 + 20G\eta_c^2\eta^2 \sum_{i=1}^{M} \frac{m_i^2\alpha_i}{m^2 n_i} I_i\epsilon^2 + 20G\eta_c^2\eta^2 \sum_{i=1}^{M} \frac{m_i^2\alpha_i}{m^2 n_i} I_i\|\nabla f(\bar{\mathbf{x}}^t)\|^2. \tag{59}$$

Substituting Eq. 24 and Eq. 59 back into Eq. 30 and rearranging the order, we have

$$\eta_g\eta_c\eta G\Big(\frac{1}{2} - 40G^2\eta_c^2\eta^2 L^2 - 100\eta^2 L^2 \sum_{i=1}^{M} \frac{m_i}{m} I_i^2 - 10\eta_g\eta_c\eta L \sum_{i=1}^{M} \frac{m_i^2\alpha_i}{m^2 n_i} I_i\Big)\|\nabla f(\bar{\mathbf{x}}^t)\|^2$$

$$\overset{(a)}{\le} f(\bar{\mathbf{x}}^t) - \mathbb{E}_t[f(\bar{\mathbf{x}}^{t+1})] + 9G^2\eta_g\eta_c^3\eta^3 L^2 \frac{M}{m}\sigma^2 + 8G\eta_g\eta_c\eta^3 L^2 \sum_{i=1}^{M} \frac{m_i}{m} I_i\sigma^2 + 40G^3\eta_g\eta_c^3\eta^3 L^2\epsilon^2$$

$$+ 75G\eta_g\eta_c\eta^3 L^2 \sum_{i=1}^{M} \frac{m_i}{m} I_i^2\epsilon_i^2 + 100G\eta_g\eta_c\eta^3 L^2 \sum_{i=1}^{M} \frac{m_i}{m} I_i^2\epsilon^2 + \frac{1}{2}G\eta_g^2\eta_c^2\eta^2 L \sum_{i=1}^{M} \frac{m_i^2}{m^2 n_i}\sigma^2$$

$$+ \frac{3}{4}G\eta_g^2\eta_c^2\eta^2 L \sum_{i=1}^{M} \frac{m_i^2\alpha_i}{m^2 n_i}\sigma^2 + \frac{15}{2}G\eta_g^2\eta_c^2\eta^2 L \sum_{i=1}^{M} \frac{m_i^2\alpha_i}{m^2 n_i} I_i\epsilon_i^2 + 10G\eta_g^2\eta_c^2\eta^2 L \sum_{i=1}^{M} \frac{m_i^2\alpha_i}{m^2 n_i} I_i\epsilon^2. \tag{60}$$

Likewise, we drop the term $\mathbb{E}_t\|\frac{1}{m}\sum_{i=1}^{M}\sum_{\tau=0}^{\omega_i-1}\sum_{h=0}^{I_i-1}\sum_{j\in\mathcal{V}_i}\nabla F_j(\mathbf{x}_j^{t,\tau,h})\|^2$ with condition $\eta_g\eta_c\eta \le \frac{1}{GL}$.

Suppose $40G^2\eta_c^2\eta^2 L^2 + 100\eta^2 L^2\sum_{i=1}^{M}\frac{m_i}{m}I_i^2 + 10\eta_g\eta_c\eta L\sum_{i=1}^{M}\frac{m_i^2\alpha_i}{m^2 n_i}I_i < \frac{1}{2}$, and there exists a constant $c > 0$ satisfying $(\frac{1}{2} - 40G^2\eta_c^2\eta^2 L^2 - 100\eta^2 L^2\sum_{i=1}^{M}\frac{m_i}{m}I_i^2 - 10\eta_g\eta_c\eta L\sum_{i=1}^{M}\frac{m_i^2\alpha_i}{m^2 n_i}I_i) > c > 0$, then we have

$$\|\nabla f(\bar{\mathbf{x}}^t)\|^2 \le \frac{f(\bar{\mathbf{x}}^t) - \mathbb{E}_t[f(\bar{\mathbf{x}}^{t+1})]}{\eta_g\eta_c\eta Gc} + \frac{1}{c}\Big[9G\eta_c^2\eta^2 L^2 \frac{M}{m}\sigma^2 + 8\eta^2 L^2 \sum_{i=1}^{M} \frac{m_i}{m} I_i\sigma^2$$

$$+ 75\eta^2 L^2 \sum_{i=1}^{M} \frac{m_i}{m} I_i^2\epsilon_i^2 + 100\eta^2 L^2 \sum_{i=1}^{M} \frac{m_i}{m} I_i^2\epsilon^2 + 40G^2\eta_c^2\eta^2 L^2\epsilon^2 + \frac{1}{2}\eta_g\eta_c\eta L \sum_{i=1}^{M} \frac{m_i^2}{m^2 n_i}\sigma^2$$

$$+ \frac{3}{4}\eta_g\eta_c\eta L \sum_{i=1}^{M} \frac{m_i^2\alpha_i}{m^2 n_i}\sigma^2 + \frac{15}{2}\eta_g\eta_c\eta L \sum_{i=1}^{M} \frac{m_i^2\alpha_i}{m^2 n_i} I_i\epsilon_i^2 + 10\eta_g\eta_c\eta L \sum_{i=1}^{M} \frac{m_i^2\alpha_i}{m^2 n_i} I_i\epsilon^2\Big], \tag{61}$$

Taking a double expectation over the data samples among all workers and averaging from $t = 0, 1, \ldots, T$, we have the final results as

$$\min_{t\in[T]} \mathbb{E}\|\nabla f(\bar{\mathbf{x}}^t)\|^2 \le \frac{f_0 - f_*}{c\eta_g\eta_c\eta GT} + \frac{1}{c}(\Phi_1 + \Phi_2 + \Phi_3), \tag{62}$$

where,

$$\Phi_1 = 9G\eta_c^2\eta^2 L^2 \frac{M}{m}\sigma^2 + 8\eta^2 L^2 \sum_{i=1}^{M} \frac{m_i}{m} I_i\sigma^2 + 40G^2\eta_c^2\eta^2 L^2\epsilon^2$$

$$+ 75\eta^2 L^2 \sum_{i=1}^{M} \frac{m_i}{m} I_i^2\epsilon_i^2 + 100\eta^2 L^2 \sum_{i=1}^{M} \frac{m_i}{m} I_i^2\epsilon^2, \quad \Phi_2 = \frac{1}{2}\eta_g\eta_c\eta L \sum_{i=1}^{M} \frac{m_i^2}{m^2 n_i}\sigma^2,$$

$$\Phi_3 = \frac{3}{4}\eta_g\eta_c\eta L \sum_{i=1}^{M} \frac{m_i^2\alpha_i}{m^2 n_i}\sigma^2 + \frac{15}{2}\eta_g\eta_c\eta L \sum_{i=1}^{M} \frac{m_i^2\alpha_i}{m^2 n_i} I_i\epsilon_i^2 + 10\eta_g\eta_c\eta L \sum_{i=1}^{M} \frac{m_i^2\alpha_i}{m^2 n_i} I_i\epsilon^2. \tag{63}$$

This completes the proof of Theorem 2. $\qquad\square$

# E  ADDITIONAL RESULTS

## E.1  WEAKENING EFFECT ON CIFAR-10

We conduct similar experiments on CIFAR-10 to show the weakening effect of partial participation for HFL. Here we only test the partial worker case with $20\%$ participating.

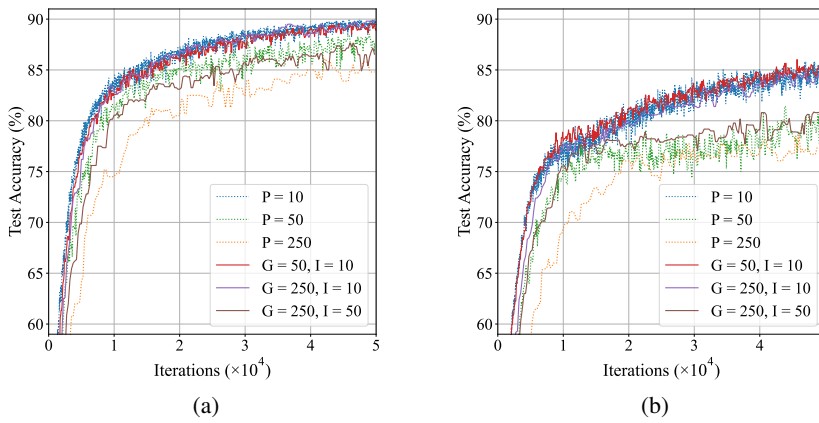

(a)                                                                                (b)

Figure 2: Test Accuracy w.r.t. iterations on CIFAR-10. (a) Group i.i.d. with partial participation; (b) Group non-i.i.d. with partial participation.

Similarly, we observe that the curve patterns in Fig. 2(b) resembles those in Fig. 2(a), namely there is no noticeable performance gap (e.g., $G = 50, I = 10$ and $G = 250, I = 10$ versus $P = 10$). Note that HFL with $G = 250, I = 50$ even performs slightly better than standard FL with $P = 50$.

## E.2 SAMPLING STRATEGY

We explore the performance of different sampling strategies on MNIST. Here we use two uniform groups both with 50 workers. The round periods are set to $G = 100, I = 50$.

Table 7: Master rounds needed for sampling strategies to achieve target test accuracy on MNIST.

| Sampling Ratio | 10% | 20% | 30% | 40% | 50% | 60% | 70% | 80% | 90% | 100% |
|---|---|---|---|---|---|---|---|---|---|---|
| Strategy 1 | 643 | 491 | 492 | 469 | 444 | 444 | 431 | 438 | 409 | 441 |
| Strategy 2 | 564 | 485 | 477 | 414 | 414 | 411 | 411 | 399 | 379 | 377 |

Table 7 presents the results for different sampling strategies. We observe that strategy 2 always outperform strategy 1 with the same sampling ratio. This matches our Theorem 2 as convergence bound for strategy 2 is tighter due to the additional coefficient $\alpha_i \leq 1, \forall i \in [M]$. Basically, the results indicates a trend that the higher the sampling ratio, the better the convergence. Still, note that strategy 1 with $100\%$ shows a little performance degradation. This may be a high sampling ratio with replacement may include multiple stochastic gradient updates from a single worker, which instead potentially leads to higher variance.