# OpenReview forum: "On the Convergence of Hierarchical Federated Learning with Partial Worker Participation"
_auai.org/UAI/2024/Conference — UAI 2024 poster_

### Official Review · Reviewer_6etD · 2024-03-21

**Q2-1 Originality-Novelty:** 3
**Q2-2 Correctness-Technical Quality:** 3
**Q2-5 Clarity Of Writing:** 4

**Q10 Ethical Concerns:**

No.

**Q1 Summary And Contributions:**

- The authors examine the problem of Hierarchical Federated Learning (HFL), which conducts additional parallel local aggregations within multiple local groups compared to classical Federated Learning (FL). This structure is able to balance the communication overhead and the learning performance.

- The authors propose an algorithm with three-sided learning rates and provide convergence analysis for the full worker participation (FWP) and the partial worker participation (PWP) settings. The convergence rate of their algorithm is tighter than previous results. They also conduct numerical experiments to demonstrate the effectiveness of their algorithm, supporting their theoretical guarantees.

**Q2-3 Extent To Which Claims Are Supported By Evidence:**

3: Good: the main claims are supported by convincing evidence (in the form of adequate experimental evaluation, proofs, (pseudo-)code, references, assumptions).

**Q2-4 Reproducibility:**

3: Good: key resources (e.g. proofs, code, data) are available and key details (e.g. proofs, experimental setup) are sufficiently well-described for competent researchers to confidently reproduce the main results.

**Q3 Main Strengths:**

- The convergence analysis part is well-organized and bridges the gap between FWP and PWP settings in HFL.

- The results of experiments support the theoretical guarantees and verify several phenomenons, e.g., "the weakening effect".

**Q4 Main Weakness:**

The idea of hierarchical structure in FL sounds interesting to me. However, I am not sure of the definition of "communication time" in the experiment section. According to Appendix D.2, you assume that worker-master RTT is ten times as the worker-cluster one. Then in my understanding of Algorithm 1, your advantage of communication overhead comes from turning parts of worker-master communication into worker-cluster one. If there is no significant difference in these communication cost, will the result of your algorithm generally equal to the standard FL as [1]?

[1] Haibo Yang, Minghong Fang, and Jia Liu. Achieving linear speedup with partial worker participation in non-iid federated learning. In International Conference on Learning Representations, 2021.

**Q5 Detailed Comments To The Authors:**

In Section 3:

In Line 12 of Algorithm 1, I think $\mathbf{x}^{t,\tau,I_i}$ should be $\mathbf{x}_j^{t,\tau,I_i}$.

In Section 5:

You do not mention the specification of parameters $\epsilon$ and $\epsilon_i$. Are they empirically evaluated? Please provide more information about your method.

**Q9 Complying With Reviewing Instructions:**

Yes

---

> ### Author Rebuttal · Authors · 2024-04-07
>
> Thank you very much for your comments.
>
> The primary contribution of our work is to provide a newly unified convergence bound for HFL settings. Our theoretical results extend the existing HFL framework to accommodate partial participation and heterogeneous clusters $I_i$ and $n_i$. By setting $M = 1, I = G$ and $\epsilon = 0$, our results align with the findings of FL from [Yang et al. 2021].
>
> If there is minimal disparity in worker-master RTT and worker-cluster RTT, the communication costs of HFL and standard FL [Yang et al. 2021] will be nearly identical. On the other hand, a critical factor driving the widespread adoption of HFL is its low latency. By distributing computational tasks across cloud servers, edge servers, and clients, data processing can occur closer to its source, reducing the latency associated with transmitting data back and forth to distant cloud servers. Therefore, there is a pressing need to examine the theoretical performance of HFL.
>
> Theoretical analysis enables us to devise a three-sided learning rates algorithm, which mitigates data divergence issues and achieves superior convergence performance. Additionally, the results suggest that adjusting the worker sampling ratio and round period to match internal divergence could potentially enhance convergence behavior, guiding the deployment of HFL.
>
> For other detailed comments, we have revised Algorithm 1.
> $\epsilon$ and $\epsilon_i$ are primarily used for theoretical analysis of convergence behavior. $\epsilon$ measures the data heterogeneity among groups, while $\epsilon_i$ measures the data heterogeneity among workers inside a group $i$. Emperically, evaluating them directly is not straightforward. We provide the implementation details regarding data partition in Appendix D.1. Concisely speaking, we control data heterogeneity by adjusting the number of class labels distribution. For instance, fewer class labels per worker or group result in greater data heterogeneity (i.e., the larger $\epsilon$ or $\epsilon_i$ is). On the contrary, randomly shuffling all data samples regardless of labels and deploying them to each worker and group, would result in an i.i.d. data distribution case (i.e., $\epsilon \approx 0$ and $\epsilon_i \approx 0$).
> This is a common setting in FL literature when studying data heterogeneity.

---

### Official Review · Reviewer_VSnn · 2024-03-22

**Q2-1 Originality-Novelty:** 3
**Q2-2 Correctness-Technical Quality:** 3
**Q2-5 Clarity Of Writing:** 3

**Q1 Summary And Contributions:**

This paper studies the convergence properties of a two-level hierarchical federated learning (HFL) problem with non-iid data in local clients, basically, local clients communicate with local clusters and clusters communicate with central servers after several local iterations at each level. The authors provide convergence analysis on the full participation and partial participation setting with explanations on why partial participation of HFL can reduce data divergences compared to conventional FL. The authors also provided numerical results to validate the theory in the end.

**Q2-3 Extent To Which Claims Are Supported By Evidence:**

3: Good: the main claims are supported by convincing evidence (in the form of adequate experimental evaluation, proofs, (pseudo-)code, references, assumptions).

**Q2-4 Reproducibility:**

3: Good: key resources (e.g. proofs, code, data) are available and key details (e.g. proofs, experimental setup) are sufficiently well-described for competent researchers to confidently reproduce the main results.

**Q3 Main Strengths:**

1. The overall structure is easy to follow and the main contributions are clearly presented
2. The explanations on the comparisons with baselines and the assumptions are clear
3. Theoretical results seem to be sufficient

**Q4 Main Weakness:**

1. There seems to be no discussion on designing the hierarchical structure and the clusters are treated as given, but there could be flexibility in actual production, and guidelines on designing a good hierarchical structure will be helpful
2. There seems to be no discussion on why more levels of the hierarchy are not considered, and whether or when they can bring additional benefit
3. How this work compares with decentralized FL (fully decentralized as well as partially decentralized with local servers) is not obvious.

**Q5 Detailed Comments To The Authors:**

Please see the above. A notation table is a good-to-have in the appendix.

**Q9 Complying With Reviewing Instructions:**

Yes

---

> ### Author Rebuttal · Authors · 2024-04-07
>
> Thank you very much for your comments. Our response to your concerns are as follows.
>
> 1. On one hand, for edge-based FL, clients within the communication range of the server collaborate to train a machine learning model. Generally, the location and communication range of edge servers (such as base stations) are fixed. Treating the clients within the communication range of an edge server as one cluster is a natural choice. Therefore, in this paper, we employ given clusters (i.e., assuming an arbitrary grouping) which is practical in real-world scenarios. Regarding sophisticated hierarchical structure design (such as edge server deployment, grouping strategies), it is important and worth researching, but it is not the primary task of this paper.
> On the other hand, our theoretical framework allow certain flexibility and individualization for HFL. This includes the master round $G$ for the whole learning system, different worker number $m_i$, sampling number $n_i$, and cluster period $I_i$ for each cluster. Still, different cluster may possess inherent characteristics, such as data heterogeneity. We discuss this in subsection **Overcome Divergence** of Section 4.2, where each cluster can adjust its worker sampling ratio and round period accordingly to deal with its inner data heterogeneity. We also verify this with experiments in Section 5.3.
> We intend to provide further discussions about HFL system design in our revised paper.
>
> 2. The cloud-edge-end architecture is prevalent in edge computing, constituting a two-level architecture. Therefore, our study delves into the convergence performance of two-level HFL in this paper. For HFL with more than $2$ levels, we assume that the global server is at level $\mathcal{l} = 0$. At level $\mathcal{l} = 1, 2,\ldots,N-1$, each server at level $\mathcal{l}$ connects to $M_{\mathcal{l-1}}$ servers at level $\mathcal{l-1}$. At $\mathcal{l} = N$, each server at level $N-1$ connects to $M_N$ clients. Intuitively, we can straightforwardly extend our results to multi-level HFL. The weakening effects will still apply to each connected level in multi-level HFL with PWP, aiding in reducing data divergences. We intend to provide further discussions about multi-level HFL in our revised paper.
>
> 3. Decentralized FL (fully decentralized as well as partially decentralized with local servers) holds significant promise for FL.  However, the architecture of decentralized FL differs greatly from HFL. A fair comparison between HFL and fully decentralized FL may not be straightforward. In contrast, we have opted to compare our results with the partially decentralized FL algorithm MLL-SGD [1], where it is still a two-level architecture while clusters are organized as a p2p network.
>
> We maintain the same worker learning rate $\eta=0.01$ for both HFL and MLL-SGD. For ease of comparison, we set a special case $\eta_c=1$ and $\eta_g=1$ of our algorithm, which can also be considered as a natural generalization of FedAvg in HFL (referred to as HierFedAvg).
>
> | | Iterations ($\times 10^4$) | Communication Time |
> | ---- | ---- | ---- |
> | MLL-SGD (fully-connected) | 2.23 | 2.43 |
> | MLL-SGD (ring) | 3.1 | 3.38 |
> | HierFedAvg | 2.22 | 2.42 |
>
> For MLL-SGD algorithm, we observe that its fully-connected variant is essentially equivalent to the HFL architecture. The only difference is that the aggregation among clusters is achieved through peer-to-peer communication rather than through the coordination of a master node. This can be verified by the very close performance between MLL-SGD (fully-connected) and HierFedAvg. When in a ring topology, MLL-SGD exhibits relatively slower convergence rate. This is because the model aggregation among clusters cannot always be fully synchronized.
>
> For other detailed comments, we would like to add a notation table in the appendix in the revised paper.
>
> [1] Multi- level local sgd: Distributed sgd for heterogeneous hierarchical networks. ICLR 2020.

---

### Official Review · Reviewer_5aj1 · 2024-03-22

**Q2-1 Originality-Novelty:** 2
**Q2-2 Correctness-Technical Quality:** 3
**Q2-5 Clarity Of Writing:** 3

**Q10 Ethical Concerns:**

No.

**Q1 Summary And Contributions:**

This paper studies the Hierarchical federated learning (HFL) without full work participation and i.i.d. datasets. The authors propose a unified framework for HFL covering full and partial worker participation with non-i.i.d data, non-convex objective function and stochastic gradient descent. The key of the designed algorithm is to use three learning rates to migrate the data divergence, realizing better convergence rates. Finally, experiments are conducted to validate the theoretical results.

**Q2-3 Extent To Which Claims Are Supported By Evidence:**

3: Good: the main claims are supported by convincing evidence (in the form of adequate experimental evaluation, proofs, (pseudo-)code, references, assumptions).

**Q2-4 Reproducibility:**

3: Good: key resources (e.g. proofs, code, data) are available and key details (e.g. proofs, experimental setup) are sufficiently well-described for competent researchers to confidently reproduce the main results.

**Q3 Main Strengths:**

1. The paper considers a more general setting compared with existing works, covering non-i.i.d data and partial participation
2. The unified three-sided learning rates approach gives good improvements on the convergence rate.
3. The experiments are extensive and verify the theoretical results.

**Q4 Main Weakness:**

1. The work seems a little bit incremental. The authors use existing algorithmic components in FL and the main contribution is to bring these components to the HFL setting. In Table 1, it seems strange that using SGD is one new feature that is involved in the comparison, which is quite common in the literature.
2. For the analysis, the improvements are good, but it does not seem very challenging to achieve these results. Again, the main challenge is to see if existing FWP and PWP can migrate to the setting of HFL.
3. As mentioned in the abstract, HFL is good at protecting privacy and achieving low communication costs, however, I do not see any discussion about how the current work protects privacy and about the communication cost in theory.

**Q5 Detailed Comments To The Authors:**

Please give more elaboration on the aspect of privacy protection and the communication cost.

**Q9 Complying With Reviewing Instructions:**

Yes

---

> ### Author Rebuttal · Authors · 2024-04-07
>
> Thank you very much for your detailed comments and questions. Our response to your questions are as follows.
>
> 1. We acknowledge that SGD is quite common in FL. The work detailed in [Liu et al. 2020] serves as the canonical reference for HFL, being the most highly cited technical literature on the subject. However, the results in [Liu et al. 2020] only consider full gradient. In Table 1, we feature SGD not to emphasize its sophistication but rather to illustrate the contrast with [Liu et al. 2020]. Even when setting the SGD noise $\sigma^2 = 0$, our results are tighter.
>
> 2. We here make a more detailed clarification for the theoretical challenge and contribution of our work.
> The major contribution of our work is to provide a newly unified convergence bound for HFL settings. Our theoretical results recover the previous results of FL from [Yang et al. 2021] by setting $M=1,I=G$, and $\epsilon=0$, and generalize existing HFL results to partial participation and heterogeneous clusters $I_i$ and $n_i$. We emphasize that our work is not the incremental modification compared to existing works (such as [Wang et al. 2022] or [Yang et al. 2021]). Compared to [Wang et al. 2022], our work only adopts the same typical HFL settings, while the convergence analysis framework is thoroughly different. Specifically, we **rederive a round-level convergence bound for both full and partial participation**, which is potentially tighter (in the sense of second term, our $\mathcal{O}(\frac{1}{T})$ versus their $\mathcal{O}(\frac{G}{T})$) and more general (aware of some server-sided optimization) than their iteration-level results. Compared to [Yang et al. 2021], our work shares a similar algorithmic approach but **differs in the analysis method**. The similarities between our theoretical results and theirs are aimed at providing better unification and insight.
> We do encounter several theoretical challenges specific to the HFL scenario, which have not been addressed in existing FL works. We have developed new ideas to tackle these challenges, which can be summarized into threefold, i.e., mutual bounding, uncertainty redirection, and partial-aware WC-MSE and CM-MSE. In view of this, we believe that our paper has its reference value to the community. Due to space limitation, we present the elaboration about the theoretical challenges and our solutions in a subsequent official comment. Thanks for your patience and attention.
>
> 3. HFL enables the local training of subsets of data at clients without the need to share raw data across the network, thereby preserving data privacy by minimizing the exposure of sensitive information. HFL leverages local aggregation at the edge server to merge model updates from clients before transmitting aggregated updates to the global server. This reduces the frequency of information exchange between clients and the global server, thereby lowing communication latency and enhancing scalability while still enabling global model convergence. Additionally, HFL can incorporate techniques such as differential privacy and secure aggregation to further enhance privacy protection. Moreover, we can consider gradient quantization and sparsification methods in HFL to reduce the amount of information exchange, thereby further reducing communication latency. In general, these aforementioned techniques and our work are orthogonal and can be combined together. In Appendix D.2, we provide implementation details regarding communication time. We would like to provide more discussions about privacy protection and communication costs in the revised paper.

---

### Official Review · Reviewer_B7Ub · 2024-03-26

**Q2-1 Originality-Novelty:** 2
**Q2-2 Correctness-Technical Quality:** 3
**Q2-5 Clarity Of Writing:** 3

**Q1 Summary And Contributions:**

This paper presents a unified convergence analysis framework for Hierarchical Federated Learning (HFL), encompassing scenarios of full and partial worker participation under non-i.i.d. data in non-convex objective functions. The analytical framework offers valuable insights: the local aggregation of the HFL can weaken the impacts of the worker-cluster part of the global divergence in the full participation case, and it can further weaken the worker-cluster and cluster-master part.

**Q2-3 Extent To Which Claims Are Supported By Evidence:**

3: Good: the main claims are supported by convincing evidence (in the form of adequate experimental evaluation, proofs, (pseudo-)code, references, assumptions).

**Q2-4 Reproducibility:**

3: Good: key resources (e.g. proofs, code, data) are available and key details (e.g. proofs, experimental setup) are sufficiently well-described for competent researchers to confidently reproduce the main results.

**Q3 Main Strengths:**

1. The focus of this paper is organic and follows a logical narrative in tandem with prior research (specifically, extending HFL to include full/partial client participation, etc.).

2. The convergence analysis is thorough, addressing a specific case and providing a comprehensive solution, while the discourse on mitigating divergence offers valuable insights.

3. Furthermore, the inclusion of a non-convex objective function broadens the scope beyond many existing studies, contributing to a more encompassing understanding of the subject matter.

**Q4 Main Weakness:**

1. Extending the current two-sided learning rates in Federated Learning (FL) to Hierarchical Federated Learning (HFL) with three-sided learning rates is a natural progression. However, the technical challenges associated with this extension are not explicitly outlined in the paper. As a result, the modifications made to both algorithm design and theoretical analysis are incremental in nature, as presented in the paper.

2. As highlighted by the author in the introduction, HFL achieves a delicate balance between communication overhead and learning performance. Introducing flexible communication methods tailored to the unique hierarchical structure of HFL could effectively address communication bottlenecks. For instance, leveraging asynchronous communication methods, such as those explored in [Asynchronous Federated Optimization; Anarchic Federated Learning], demonstrates promising avenues for improvement.

**Q5 Detailed Comments To The Authors:**

See Q4

**Q9 Complying With Reviewing Instructions:**

Yes

---

> ### Author Rebuttal · Authors · 2024-04-07
>
> Thank you very much for your detailed comments and questions. Our response to your questions are as follows.
>
> 1. We have actually discussed several technical difficulties and challenges in the proof sketch in our main paper. We here make a more detailed clarification.
> The primary contribution of our work is to provide a newly unified convergence bound for HFL settings. Our theoretical results recover the previous results of FL from [Yang et al. 2021] by setting $M=1,I=G$, and $\epsilon=0$, and generalize existing HFL results to partial participation and heterogeneous clusters $I_i$ and $n_i$. We emphasize that our work is not the incremental modification compared to existing works (such as [Wang et al. 2022] or [Yang et al. 2021]). Compared to [Wang et al. 2022], our work only adopts the same typical HFL settings, while the convergence analysis framework is thoroughly different. Specifically, we **rederive a round-level convergence bound for both full and partial participation**, which is potentially tighter (in the sense of second term, our $\mathcal{O}(\frac{1}{T})$ versus their $\mathcal{O}(\frac{G}{T})$) and more general (aware of some server-sided optimization) than their iteration-level results. Compared to [Yang et al. 2021], our work shares a similar algorithmic approach but **differs in the analysis method**. The similarities between our theoretical results and theirs are aimed at providing better unification and insight.
> We do encounter several theoretical challenges specific to the HFL scenario, which have not been addressed in existing FL works. We have developed new ideas to tackle these challenges, which can be summarized into threefold, i.e., mutual bounding, uncertainty redirection, and partial-aware WC-MSE and CM-MSE. In view of this, we believe that our paper has its reference value to the community. Due to space limitation, we present the elaboration about the theoretical challenges and our solutions in a subsequent official comment. Thanks for your patience and attention.
>
> 2. It is true that more flexible communication protocols, such as asynchronous mode, hold great potential for HFL. However, to pave the way for HFL, our work primarily aims to provide a comprehensive and unified convergence framework for the hierarchical architecture under general settings. This marks a significant initial stride. Therefore, in this paper, we concentrate on elucidating the advantages of the hierarchical architecture in HFL. For explorations on HFL with more flexible communication designation and corresponding theoretical analysis, we would like to leave to our future works.

---

### Official Review · Reviewer_SPWU · 2024-03-28

**Q2-1 Originality-Novelty:** 3
**Q2-2 Correctness-Technical Quality:** 4
**Q2-5 Clarity Of Writing:** 4

**Q1 Summary And Contributions:**

The paper presents a novel theoretical and experimental analysis for Federated Learning (HFL) that incorporate a two-level communication network architecture. It addresses the limitations of existing studies by considering scenarios with partial/full worker participation and non-i.i.d. datasets.

**Q2-3 Extent To Which Claims Are Supported By Evidence:**

2: Fair: the main claims are somewhat supported by evidence (but the experimental evaluation may be weak, or does not match entirely with the claims, important baselines may be missing, proofs contain important ideas but lack rigor, algorithmic details are only discussed superficially, references are imprecise, assumptions are not sufficiently motivated or explicated, etc.).

**Q2-4 Reproducibility:**

2: Fair: key resources (e.g. proofs, code, data) are unavailable but key details (e.g. proof sketches, experimental setup) are sufficiently well-described for an expert to confidently reproduce the main results.

**Q3 Main Strengths:**

1. The paper proposes a novel framework for HFL that accounts for partial worker/full worker participation, with non-i.i.d. data and non-convex objective function.
2. Algorithm with three-sided learning rates that is demonstrated to improves convergence performance for non-iid data.
3. The outcomes show that PWP can significantly reduce data divergences compared to standard FL setting.
4. Detailed convergence analysis for both FWP and PWP and extensive experiments that confirm the theoretical results

**Q4 Main Weakness:**

1. Tuning three learning rates specific for each problem and client settings is challenging and raises concerns with large number of clients (scalability concern)
2. There are several works on Clustered FL itself and several on hierarchical FL. The paper doesn't consider any SOTA methods for baseline comparison which is crucial given the available methods already.

**Q5 Detailed Comments To The Authors:**

1. There are few minor typos in this paper. I recommend the authors to proof-read this paper again (Grammarly is a suggestion).
	1. "sandwich" vs "sandwitch"?
2. Please maintain same "Y-axis" scale between iid and non-iid plots. They look similar but they are not if the scales are same which will make it obvious.
3. Eqn 9 in the appendix. Did you use Young's inequality (alpha=1) before applying Assumption 1? I believe. If so please add it in the for readers to understand it.
4. Perhaps compare the existing work with and without communication noise?
    a. Towards flexible device participation in federated learning - AISTAT 2021.
    b. On the Convergence of Decentralized Federated Learning Under Imperfect Information Sharing - CDC 2023

**Q9 Complying With Reviewing Instructions:**

Yes

---

> ### Author Rebuttal · Authors · 2024-04-07
>
> Thank you very much for your comments and suggestions. Our response to your questions is as follows.
>
> 1. In Corollary 1, 2, and 3, we propose the recomended configurations of three-sided learning rates for general HFL settings. Theoritically, these configurations can gurantee the desired convergence rate $\mathcal{O}(\frac{1}{\sqrt{mGT}} + \frac{1}{T})$ (or $\mathcal{O}(\frac{I_{max}}{\sqrt{m\kappa_{min}GT}} + \frac{1}{T})$) without any scalability concern. We also discuss additional choices of hyperparameters for different client settings in subsection **Overcome Divergence** of Section 4.2. In practice, due to engineering considerations, there may be issues of feasibility or scalability in setting the learning rates for HFL. We believe our convergence results can serve as theoretical guidance for the engineering design of HFL systems.
>
> 2. We have conducted experiments with several SOTA methods in HFL literature. Due to space limitation, we exclusively present the results and discussions in a subsequent official comment. We have published our code in an anonymous repository https://anonymous.4open.science/r/HFL-7F22.
>
> For other detailed comments on the writing, we have fixed our typos and aligned the ``Y-axis'' plots in Appendix E.1. Besides, we checked Eq 9 in the appendix, we actually use an extension of Jensen inequality as $\Vert\sum_{i=1}^{M} x_i \Vert^2 \leq M \sum_{i=1}^{M} \Vert x_i \Vert^2$ before applying Assumption 1. We have added this detail for ease of comprehension.

---

### Meta-Review · Area_Chair_7pPo · 2024-04-17

This paper contributes to the field of hierarchical federated learning by
introducing a novel framework that effectively handles partial worker
participation and non-i.i.d data conditions. The authors have developed a
sophisticated theoretical model and supported their claims with comprehensive
mathematical proofs and extensive experimental results.


Pros:
+ The paper presents a significant theoretical contribution by extending the
  hierarchical federated learning (HFL) to accommodate partial worker
  participation and non-i.i.d. datasets, which is a substantial advance over
  existing methodologies.
+ Extensive experiments are conducted that support the theoretical findings and
  demonstrate the effectiveness of the proposed methods in various settings.
+ The paper is technically robust, with careful mathematical derivations and
  convergence analysis that are well-supported by rigorous proofs.
+ Despite some resources being unavailable, the paper provides sufficient
  details for an expert in the field to reproduce the main results. The writing
  is clear and well-organized, making complex concepts accessible.

Cons:

- The method involves tuning three learning rates, which could be challenging in
  practice, especially with a large number of clients, potentially affecting
  scalability.
- The paper does not adequately compare its approach against state-of-the-art
  methods in clustered and hierarchical FL, which is crucial for establishing the
  efficacy of the proposed methods.
- Some claims are not fully supported by the experimental evaluation.

The paper's impact is somewhat diminished by the absence of direct comparisons
with state-of-the-art methods, which would have more concretely positioned this
work within the existing research landscape. Additionally, the practical
application of the proposed methods could be hindered by the complexity of
tuning multiple learning rates, especially in scenarios involving many clients,
which the authors acknowledge but do not fully address.


I recommend the acceptance of this paper. However, I trust that the authors will
 incorporate the suggestions provided in the reviews to improve the quality of
 the paper. In particular, the need for more comprehensive baseline comparisons
 and a deeper discussion on the practical implementation challenges should be
 included.